https://doi.org/10.1038/s41467-021-22653-8　　**OPEN**

# Learning a genome-wide score of human–mouse conservation at the functional genomics level

Soo Bin Kwon [ID] [1,2] & Jason Ernst [ID] [1,2,3,4,5,6,7 ✉]

Identifying genomic regions with functional genomic properties that are conserved between human and mouse is an important challenge in the context of mouse model studies. To address this, we develop a method to learn a score of evidence of conservation at the functional genomics level by integrating information from a compendium of epigenomic, transcription factor binding, and transcriptomic data from human and mouse. The method, Learning Evidence of Conservation from Integrated Functional genomic annotations (LECIF), trains neural networks to generate this score for the human and mouse genomes. The resulting LECIF score highlights human and mouse regions with shared functional genomic properties and captures correspondence of biologically similar human and mouse annotations. Analysis with independent datasets shows the score also highlights loci associated with similar phenotypes in both species. LECIF will be a resource for mouse model studies by identifying loci whose functional genomic properties are likely conserved.

[1] Bioinformatics Interdepartmental Program, University of California, Los Angeles, CA, USA. [2] Department of Biological Chemistry, University of California, Los Angeles, CA, USA. [3] Eli and Edythe Broad Center of Regenerative Medicine and Stem Cell Research at University of California, Los Angeles,  CA, USA. [4] Computer Science Department, University of California, Los Angeles, CA, USA. [5] Department of Computational Medicine, University of California, Los Angeles, CA, USA. [6] Jonsson Comprehensive Cancer Center, University of California, Los Angeles, CA, USA. [7] Molecular Biology Institute, University of California, Los Angeles, CA, USA. ✉email: jason.ernst@ucla.edu

Many studies interrogate human loci of interest, such as those implicated in genome-wide association studies (GWAS), by perturbing their homologous loci in mouse[1–4]. A key question in this context is the extent to which the homologous loci in mouse is expected to have similar roles to the human loci. Conversely, loci associated with phenotypes can be discovered in mouse first, raising the question of the degree to which their properties are shared with human[5].

A relatively large percentage of the human genome, ~40%, has a homologous locus in the mouse genome as determined by human–mouse pairwise sequence alignment[6]. However, a much smaller fraction of bases in these aligning pairs of loci are constrained at the sequence level[7–10]. This is because many bases are within regions whose sequences are similar enough to be aligned between species, but not necessarily constrained, which is defined at a higher resolution and generally has even greater sequence similarity. In general, it is unclear to what extent human and mouse loci that align to each other have similar properties, in particular, functional genomic properties. With large-scale functional genomic resources of genome-wide maps of chromatin accessibility, transcription factor (TF) binding, histone modifications, gene expression data across diverse cell and tissue types that have become available in mouse[11,12] in addition to human[13–15], there is an opportunity to systematically and confidently detect evidence of conservation at the functional genomics level between these species.

Previous work comparing cross-species functional genomics data to infer conservation have largely focused on comparing pairs of matched experiments for the same assay in a corresponding cell or tissue type across species[16–21]. While useful, data from a pair of experiments from two species provides limited information for differentiating evidence of conservation from similarity observed by chance. Studies that jointly compare multiple pairs of experiments from different biological conditions have additional information available for inferring conservation of functional genomic properties[17,18,20,21]. However, such approaches have often relied on manually matching corresponding experiments and have not been scaled to leverage the vast amounts of diverse data available in both human and mouse. The challenge in taking advantage of such data is that many experiments do not have an obvious corresponding experiment, and even when one is assumed there could in practice be confounding differences. Previous work partly addressed some of these issues[11,22–27], but still limited their work to one data type at a time and thus only utilized a small fraction of the available data to find evidence of conservation. Given the increasingly diverse functional genomic resources available for human and mouse, there is a need for an integrative method to better leverage those resources to infer evidence of conservation at the functional genomics level between human and mouse.

Thus, here we develop Learning Evidence of Conservation from Integrated Functional genomic annotations (LECIF), a supervised learning approach that quantifies evidence of conservation based on large-scale functional genomic data from a pair of species, which we apply to human and mouse. While LECIF leverages data from diverse cell types collected by various assays, it does not require explicit matching of experiments from different species by biological source or data type. LECIF uses pairwise sequence alignment data only to label training examples, inferring conservation from functional genomics data and not from DNA sequence. We apply LECIF to a compendium of thousands of human and mouse functional genomic annotations and learn the LECIF score for every pair of human and mouse regions that align at the sequence level. The score captures correspondence of biologically similar annotations between human and mouse, even though LECIF was not explicitly given such

information. While the LECIF score is moderately correlated with sequence constraint scores, it captures distinct information on conserved properties. The LECIF score is preferentially higher in regions previously shown to have similar phenotypic properties in human and mouse at the genetic and epigenetic level. Overall, we observe that the score can complement sequence conservation annotations in capturing human–mouse conservation and contribute to locating pairs of sequence-aligning regions whose functional genomic properties are likely conserved. We thus expect the human–mouse LECIF score will be an important resource for studies using mouse as a model organism.

## Results

**Overview of LECIF**. LECIF quantifies evidence of conservation between human and mouse genomic regions at the functional genomics level based on a large and diverse set of functional genomic annotations (Fig. 1). LECIF uses functional genomic features as input to an ensemble of neural networks, where sequence alignment information is used to label training data, but not as features ("Methods"). For training data, positive examples are pairs of human and mouse regions that align at the sequence level, while negative examples are randomly mismatched pairs of human and mouse regions that do not align to each other (Fig. 1a). All human and mouse regions included in negative examples align somewhere in the mouse and human genomes, respectively, which allows LECIF to learn pairwise characteristics of aligning human and mouse regions instead of the characteristics of regions that align to the other genome in general. LECIF assumes that positive examples are more likely to be conserved at the functional genomics level than negative examples. Since neighboring bases are likely annotated by the same annotations and for computational considerations, training examples and predictions were generated at every 50 bp within each pairwise alignment block ("Methods"). As a result, we provided the classifier with >2 million positive and >2 million negative training examples, which covered up to 90 Mb of the human and mouse genomes.

For each example, there were >8000 human and >3000 mouse functional genomic features defined. Among these features were binary features corresponding to whether a genomic base overlapped with peak calls from DNase-seq experiments, ChIP-seq experiments of TFs, histone modifications and histone variants, and cap analysis of gene expression (CAGE) experiments. In addition, there were binary features corresponding to each state and tissue combination of ChromHMM[28] chromatin state annotations and numerical features corresponding to normalized signals from RNA-seq experiments. These data covered a wide range of cell and tissue types and were generated by the ENCODE[13], Mouse ENCODE[11], Roadmap Epigenomics Project[14], or FANTOM5[29] consortia ("Methods" and Supplementary Data 1). We did not provide pairwise alignment or DNA sequence information as features to the classifier so that LECIF infers conservation specifically at the functional genomics level rather than at the sequence level.

After training, we used the classifier to make genome-wide predictions at 50 bp resolution or finer, annotating the 40% of the human genome that aligns to mouse and those aligning regions in the mouse genome with the LECIF score (Figs. 1b and 2a). We weighted negative examples 50 times more than positive examples during training because we wanted the LECIF score to highlight regions with strong evidence of conservation at the functional genomics level. As a result, a small fraction of the aligning regions was highlighted with high LECIF score, whereas most aligning regions would have scored high if the score was learned with positive and negative examples weighted equally (Fig. 2b, Supplementary Fig. 1a).

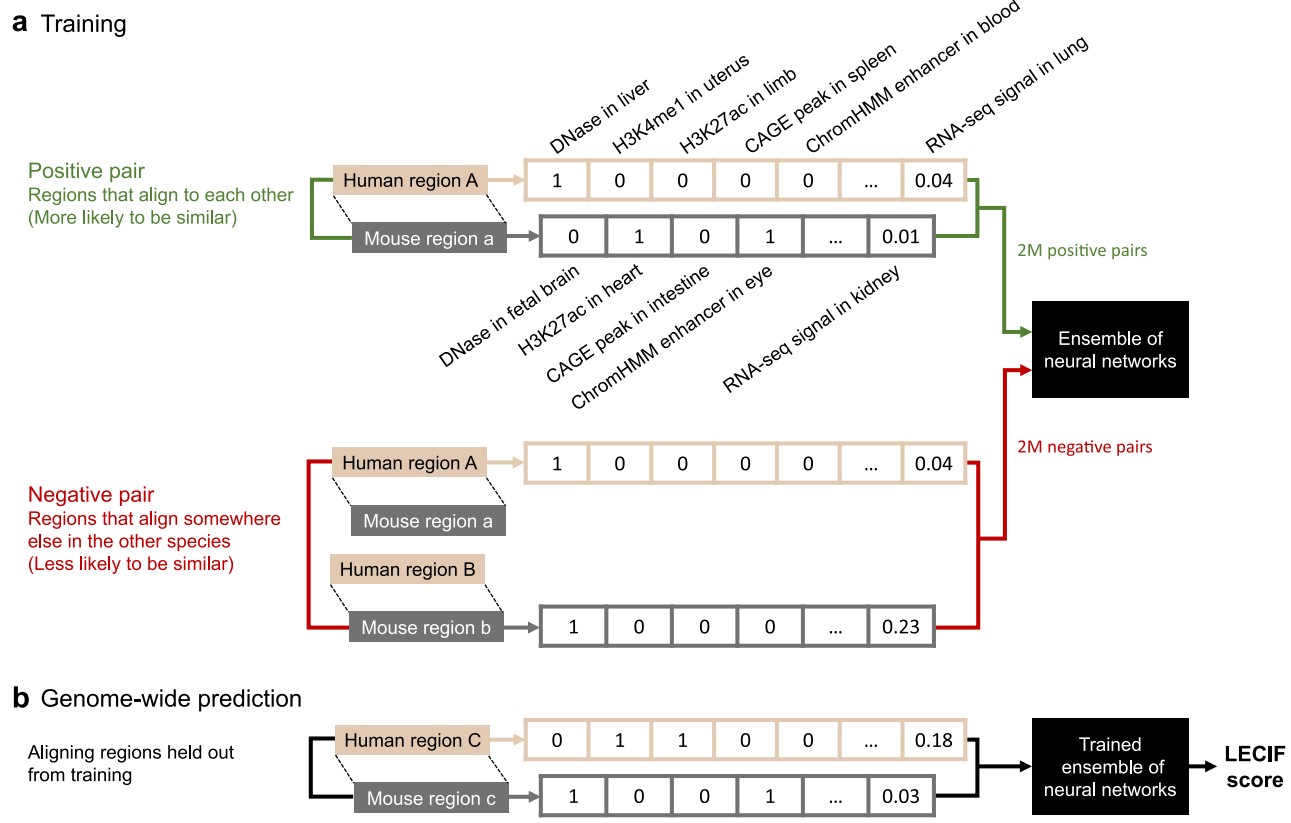

**Fig. 1 Overview of the LECIF method. a** Supervised learning procedure of LECIF. For every pair of human and mouse genomic regions, two feature vectors are generated from their functional genomic annotations, one vector for the human region (beige) and the other vector for the mouse region (gray). Each feature vector consists of thousands of functional genomic annotations, as listed in Supplementary Data 1. Only a subset of the features is shown here. These two species-specific feature vectors are given to an ensemble of neural networks (ENN). The ENN is trained to distinguish positive pairs (green), which are aligning human and mouse regions, from negative pairs (red), which are randomly mismatched human and mouse regions that do not align to each other, but somewhere else in the other species. Here, we provide about 2 million positive and 2 million negative training examples. Feature labels (e.g., DNase in liver) and matching of features across species are not provided to LECIF. **b** Genome-wide prediction procedure of LECIF. Once trained as illustrated in **a**, the ENN can estimate the probability of any given pair of human and mouse regions being classified as a positive pair. We consider this probability, the LECIF score, to represent the evidence of conservation observed in the functional genomics data annotating the given pair. Here, we generate the LECIF score for all pairs of aligning human and mouse regions. Although not shown here, for model evaluation we also generate predictions for randomly mismatched negative pairs held out from training. When generating a prediction for a pair, LECIF uses an ENN trained on data excluding the pair as described in "Methods" and Supplementary Data 2.

**Comparative evaluation of LECIF's predictive performance.** We evaluated LECIF at predicting whether pairs of regions that were held out from training align at the sequence level. LECIF had strong predictive power for this with an area under the receiver operating characteristic curve (AUROC) of 0.87 and an area under the precision-recall curve (AUPRC) of 0.23 compared to a random expectation of 0.50 and 0.02, respectively (Fig. 2c, d). In addition, scores that were trained on non-overlapping sets of chromosomes had strong agreement with each other with a Pearson correlation coefficient (PCC) of 0.90 ("Methods").

We compared LECIF to alternative methods that used random forest (RF), canonical correlation analysis (CCA), deep canonical correlation analysis (DCCA), or logistic regression (LR) instead of an ensemble of neural networks (Fig. 2c, d). When classifying held-out test examples, LECIF outperformed these methods with statistically significantly better AUROC and AUPRC values (RF AUROC: 0.82; CCA AUROC: 0.81; DCCA AUROC: 0.81; RF AUPRC: 0.13; CCA AUPRC: 0.06; DCCA AUPRC 0.07; LR AUROC: 0.50; AUPRC: 0.02; Wilcoxon signed-rank test $P <$ 0.0001). LR had no predictive power as expected, since it only considers features marginally and the positive and negative

examples were defined such that each feature has an identical marginal distribution in positive and negative data.

We next evaluated LECIF design choices by comparing the LECIF score to predictions based on alternative choices. We first compared the LECIF score with a score computed at a single-base resolution and confirmed they were strongly correlated (PCC: 0.99; "Methods"). We also compared the LECIF score to scores learned with different weightings of positive and negative examples and confirmed that relative ranking of predictions and predictive power for aligning regions were robust (Supplementary Fig. 1). We used LECIF with an ensemble of 100 neural networks and confirmed it led to better performance than using fewer networks, although fewer networks could be used to save computational cost with a small decrease in performance (Supplementary Figs. 2 and 3). We also compared the LECIF score to scores learned separately for the coding and noncoding genomes and observed that the scores were relatively well-correlated with the original LECIF score in the coding (PCC: 0.71) and noncoding (PCC: 0.95) genomes (Supplementary Fig. 4 and "Methods").

In addition, we evaluated the effect of the number of mouse features on LECIF's performance by learning two models with

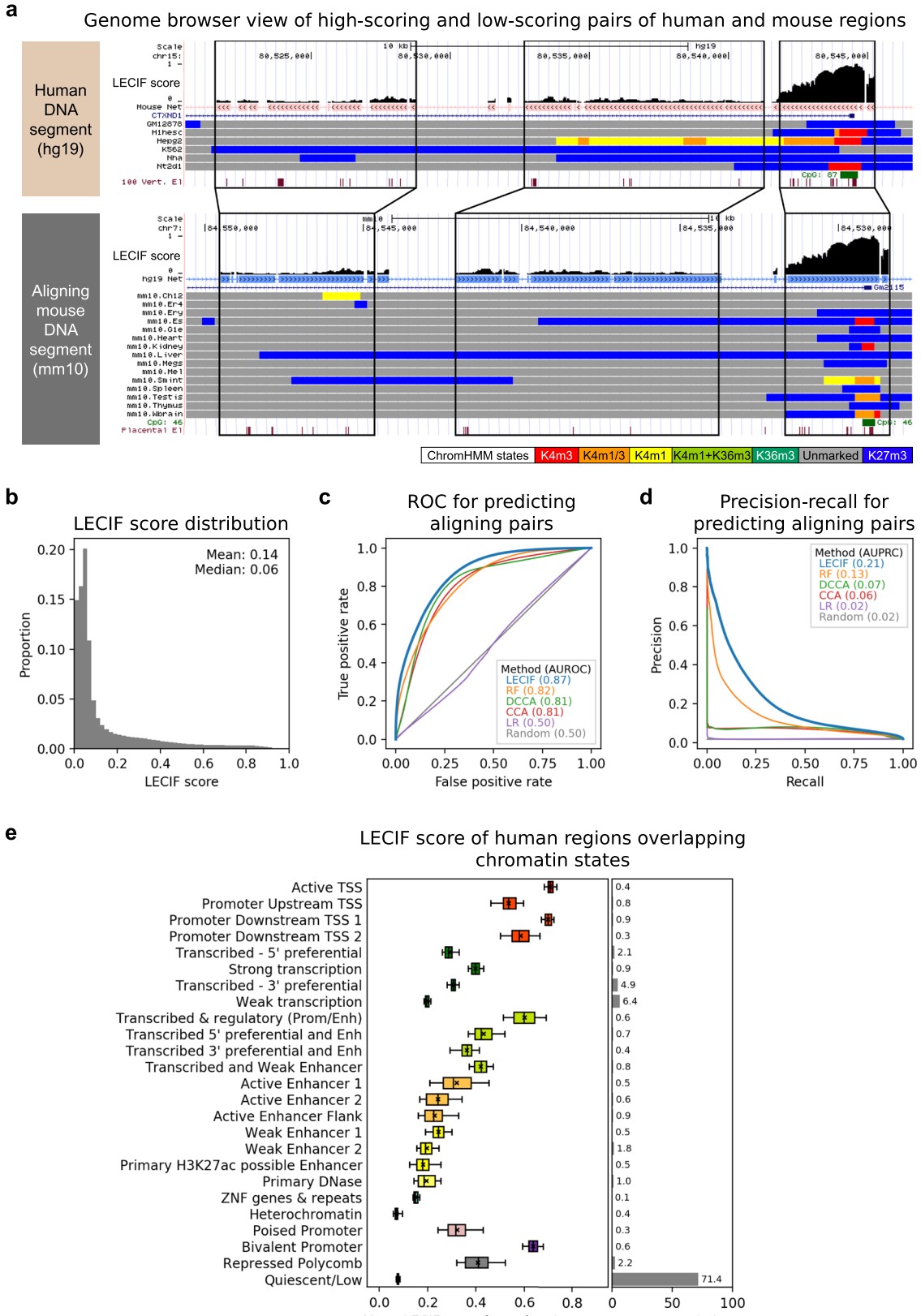

fewer mouse features ("Methods"). A score learned with 10% of the mouse features had strong agreement with the original LECIF score (PCC: 0.88; Spearman correlation coefficient (SCC): 0.80) and slightly weaker predictive performance (AUROC: 0.83 vs. 0.86; AUPRC: 0.16 vs. 0.21;

Supplementary Fig. 5). However, a score learned with 1% of the mouse features had substantially weaker agreement with the original LECIF score (PCC: 0.66; SCC: 0.18) and weaker predictive performance for aligning pairs (AUROC: 0.66; AUPRC: 0.07).

**Fig. 2 Characteristics of the human–mouse LECIF score. a** Genome Browser[45] views with the LECIF score annotating human gene *CTXND1* (top) and its mouse ortholog *Gm2115* (bottom). In each view, LECIF score is shown in the top, followed by net alignment annotation[6] marking regions that align with colored boxes. Below the net annotation are RefSeq gene annotation[65] and ChromHMM chromatin state annotations[28] for different epigenomes from a model learned jointly for human and mouse[11]. State legend is in the bottom right. Below the state annotations are CpG island and PhastCons element[7] annotations. Black lines highlight segments that largely align. The mouse genome browser view is shown in the reverse direction (3′–5′). **b** Distribution of the LECIF score. Fifty equal-width bins were used. **c** Receiver operating characteristic (ROC) curve comparing LECIF, random forest (RF), canonical correlation analysis (CCA), deep CCA (DCCA), and logistic regression (LR) for classifying pairs of regions that align at the sequence level, evaluated on a common set of held-out test data. Legend indicates color and mean area under the ROC curve (AUROC) for each method. The curve of each method was obtained by classifying 100,000 positive and 100,000 negative examples sampled with replacement from all test examples 100 times. Negative examples were weighted 50 times more than positive examples. For each method, standard deviation of the 100 AUROC values was under 0.005. **d** Similar to **c** except showing precision-recall (PR) instead of ROC. Standard deviation of the 100 area under the PR curve (AUPRC) values was under 0.005 for all methods. **e** Left panel shows for each human chromatin state as described previously[14,66] the distribution of mean LECIF score over different epigenomes ($n = 127$). Mean LECIF score for a state in an epigenome is computed by averaging the score across regions overlapping the state in the epigenome. Each distribution is represented by a boxplot with median (black vertical line), mean (black "x"), 25th and 75th percentiles (box), and 5th and 95th percentiles (whisker). Right panel shows mean coverage of each state across human regions that align to mouse. Source data are provided as a Source Data file. A mouse version of this plot is in Supplementary Fig. 10.Source Data

**Predictive power when including adjacent non-aligning mouse regions**. The LECIF method can also score pairs of human and mouse regions that do not align at the sequence level. Previous comparative studies have reported movements of regulatory elements during evolution, where homologous regulatory activity of a human region is found in a region near the aligning region in another species instead of the aligning region[16,30,31]. We thus investigated whether it is advantageous for LECIF to consider also the scores at non-aligning mouse regions proximal to the mouse region aligning to human. Specifically, for a given aligning pair of human and mouse regions, we took the maximum LECIF score from pairs consisting of the human region and any mouse region located within a window centered around the aligning mouse region ("Methods" and Supplementary Fig. 6). We varied window sizes and repeated the same AUROC evaluations for predicting aligning regions as above (Supplementary Fig. 7).

We found that as we expanded the window size, the predictive power decreased overall. We saw similar results when we repeated the evaluation with pairs stratified by the LECIF score at the aligning regions except for pairs with the lowest LECIF score (Supplementary Fig. 8). When we trained LECIF with an alternative set of negative examples selected from a genome background and repeated the evaluations ("Methods"), the expanded window still had decreased predictive power overall (Supplementary Fig. 7). These results suggested that applying LECIF to non-aligning regions would result in a substantial increase in false-positive predictions, which indicates that sequence alignment provides strong prior information in detecting evidence for conservation at the functional genomics level. Moreover, non-aligning regions in general tend to be less conserved and exhibit different properties at the functional genomics level than aligning regions on which LECIF was trained[11], making LECIF relatively less applicable to such regions. We thus focused our initial application of LECIF to aligning regions. We note that because of the resolution at which the LECIF score is defined, even without explicitly expanding the window the score may still be capturing small movements of regulatory sites, which cannot be explicitly detected in the coarse-resolution functional genomics data currently available to LECIF.

**Distribution of LECIF score in chromatin states**. To characterize DNA elements highlighted by LECIF, we investigated the distribution of the LECIF score overlapping the chromatin state annotations that were provided to LECIF as input features. When we computed the mean LECIF score for each chromatin state across epigenomes[14] (Fig. 2e and "Methods"), chromatin states associated with strong regulatory or transcriptional activity

tended to have a higher mean LECIF score than other states, with the highest of 0.71 for an active transcription start site (TSS) state and the lowest of 0.07 and 0.08 for the heterochromatin and quiescent states, respectively. Candidate enhancer states outside of transcribed regions had an intermediate mean LECIF score ranging from 0.18 to 0.32, which was lower than the mean scores of promoter associated states, 0.53–0.71, and consistent with previous findings that enhancers tend to evolve faster than promoters[19]. We also observed similar trends with other input features and external gene annotations in both human and mouse (Supplementary Figs. 9–11).

**LECIF highlights shared functional genomic activity**. To validate that the LECIF score reflects expected cross-species similarity in functional genomic features, we investigated the LECIF score in relation to human and mouse genomic annotations jointly. We first matched a subset of human and mouse ChIP-seq experiments of H3K27ac by their tissue of origin for 14 tissue type groups ("Methods"). We then quantified the cross-species similarity of the peak calls for each pair of regions jointly across the 14 tissue type groups, using a weighted Jaccard similarity coefficient ("Methods"). We saw that the LECIF score was positively correlated with the weighted Jaccard similarity coefficient (PCC: 0.45; Fig. 3a). This is despite LECIF not being given any information regarding tissue of origin of the experiments in the compendium of functional genomic annotations.

To provide further evidence that the LECIF score reflects expected cross-species similarity in functional genomic annotations, we examined the LECIF score in relation to the chromatin state annotations of pairs of human and mouse regions. We used the state annotations from a concatenated model of ChromHMM[28], where a shared set of states were learned for human and mouse[11]. For different ranges of the LECIF score, we correlated the chromatin state frequency between human and mouse across regions in that score range ("Methods"). High-scoring pairs of regions tended to be annotated with similar sets of states in human and mouse epigenomes (Fig. 3b, c and Supplementary Fig. 12). Low-scoring pairs of regions were annotated with dissimilar sets of states in human and mouse and the quiescent state more frequently than high-scoring pairs (Fig. 3b, d and Supplementary Figs. 12–14).

We also investigated the LECIF score at topologically associated domain (TAD) boundaries that were previously identified in human and mouse cell types[32], as they represent an important regulatory genomic feature not provided to LECIF. Human regions overlapping a TAD boundary in any human cell type had a mean LECIF score of 0.17 compared to the genome-

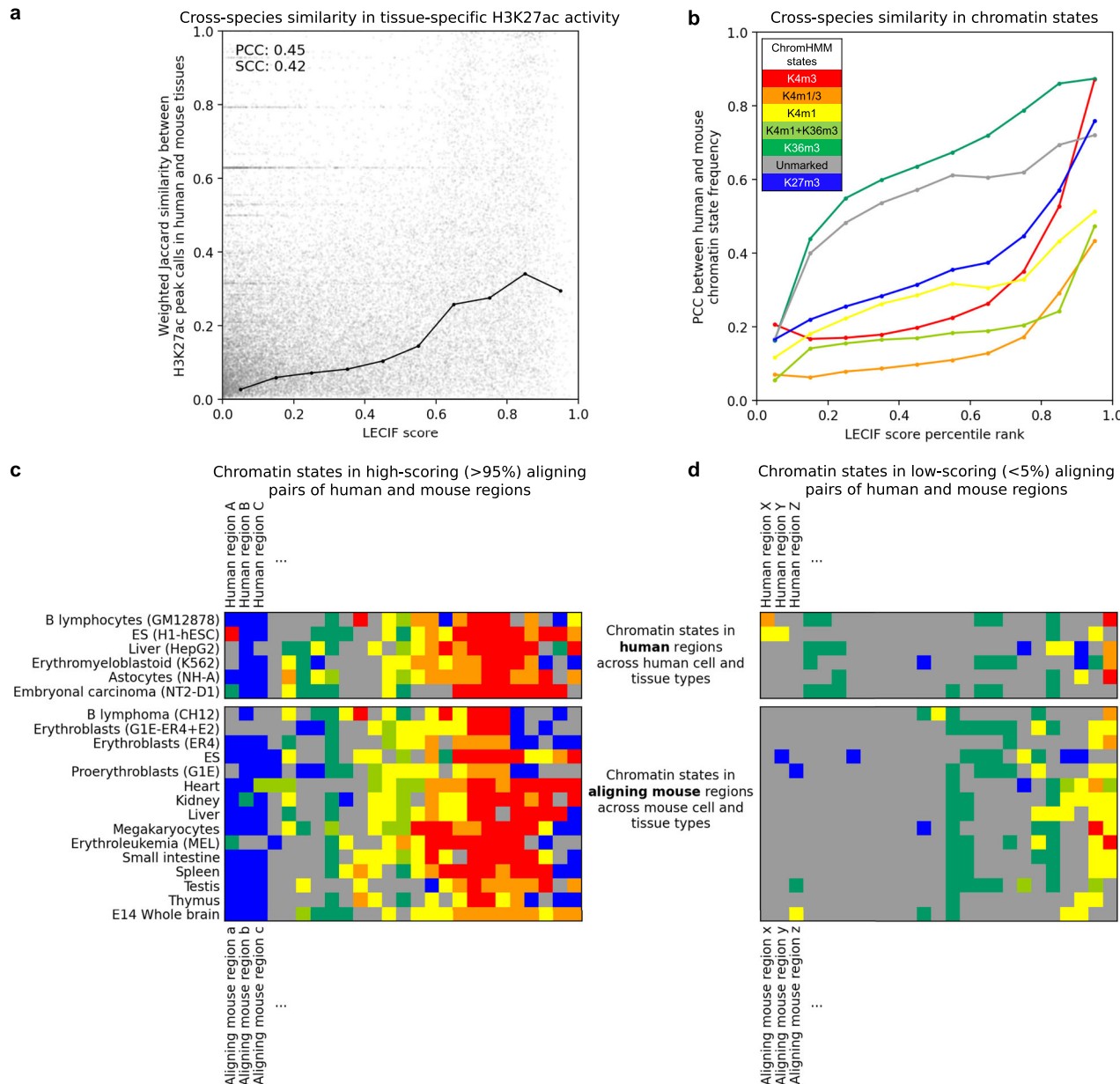

**Fig. 3 Correspondence of LECIF score to matched human and mouse annotations. a** Scatter plot showing with a gray dot for each aligning pair of human and mouse regions the LECIF score (x-axis) and cross-species similarity of H3K27ac activity (y-axis). H3K27ac activity for a region in a tissue type is quantified as the fraction of experiments in the tissue type with peak calls overlapping the region. Its cross-species similarity is quantified as the weighted Jaccard similarity coefficient over 14 matched tissue types ("Methods"). One hundred thousand random pairs are shown. PCC and SCC, computed from all regions, are shown in the top left. Black circles show the mean coefficient of pairs binned by the LECIF score using ten equal-width bins. The circles are connected by piecewise linear interpolation. Source data are provided as a Source data file. A version of this figure for the human-only baseline score is in Supplementary Fig. 16. **b** Cross-species agreement in chromatin state[11,28] frequency in aligning human and mouse regions for a ChromHMM model learned jointly for both species. Pairs were binned by LECIF score percentile rank using ten bins with similar number of pairs. For each state and percentile rank bin, we computed PCC between the human and mouse state frequencies across all pairs in the bin ("Methods"). The values are shown with circles colored according to the top left legend from ref. [11], which are connected by piecewise linear interpolation. Source data are provided as a Source data file. Alternative versions of this plot with different binning schemes are in Supplementary Fig. 12. **c** ChromHMM chromatin state[11,28] annotations in high-scoring pairs of aligning human and mouse regions. Each row in top and bottom subpanels corresponds to human and mouse epigenomes, respectively. Each column is a random pair of regions with high LECIF score (>95th percentile). Each cell shows the color of the state with which the region (column) is annotated in an epigenome (row) based on the same model as in **b**. Pairs (columns) were ordered based on hierarchical clustering applied to state annotations using Ward's linkage with optimal leaf ordering[67]. A version of this figure using mismatched non-aligning pairs is in Supplementary Fig. 17. **d** Same as **c**, but with pairs with low LECIF score (<5th percentile).Source Data.

wide mean of 0.14 (Mann–Whitney $U$ test $P < 0.0001$). Pairs with human and mouse regions both overlapping a TAD boundary in a matched cell type had an even higher mean of 0.20, scoring significantly higher than pairs with either human or mouse region or neither regions overlapping a TAD boundary in the cell type (Supplementary Fig. 15; Mann–Whitney $U$ test $P < 0.0001$).

We also verified the advantage of integrating human and mouse data by generating a human-only baseline score. The score

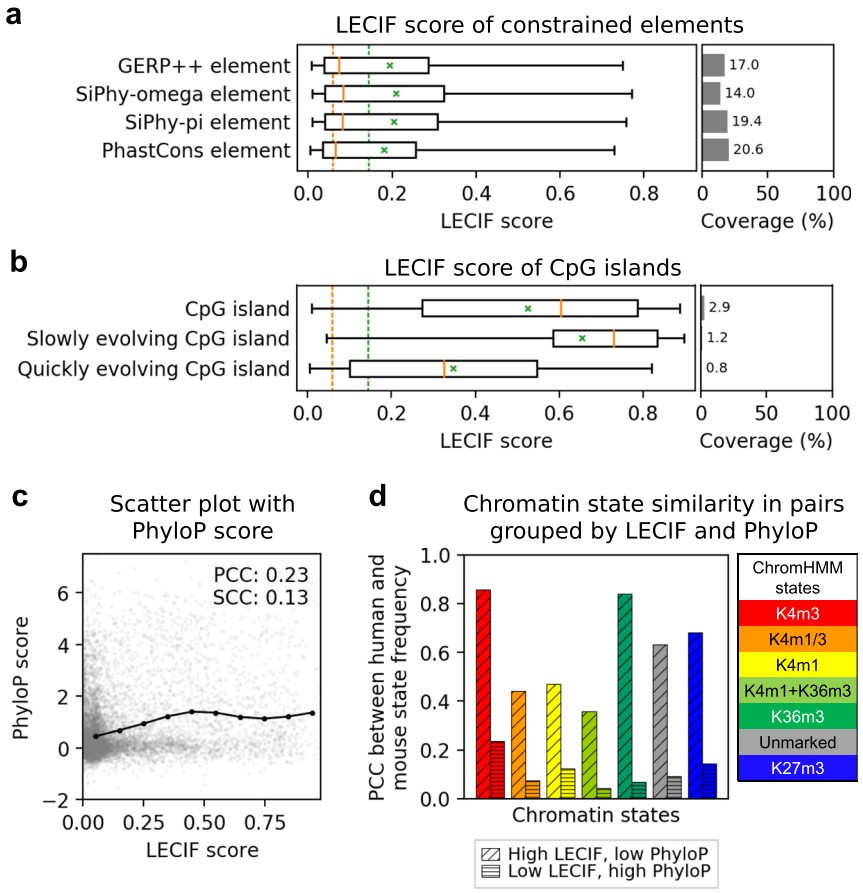

**Fig. 4 Relationship of LECIF score to sequence constraint annotations. a** Distribution of LECIF score in human regions overlapping constrained elements called by GERP++, SiPhy-omega, SiPhy-pi, and PhastCons ($n = 5,500,681, 4,515,990, 6,277,929$, and $6,634,667$ human regions, respectively)[7,9,10,68]. Each distribution is represented by a boxplot with median (orange solid line), mean (green "x"), 25th and 75th percentiles (box), and 5th and 95th percentiles (whisker). Orange and green dashed vertical lines denote genome-wide median and mean, respectively. Right subpanel shows coverage of each annotation across all human regions aligning to mouse. **b** Similar to **a**, except showing LECIF score of human regions overlapping CpG islands ($n = 950,523$), as well as subsets of regions overlapping slowly and quickly evolving CpG islands ($n = 399,280$ and $260,132$, respectively) as defined based on primates[35]. **c** Scatter plot showing with gray dots the LECIF score and PhyloP score based on a 100-way vertebrate alignment[8]. The plot displays 100,000 random human regions that align to mouse with all bases annotated by both scores. PCC and SCC, computed from all applicable regions, are shown in the top right. Mean PhyloP score of all applicable regions binned by the LECIF score with ten equal-width bins are shown in black circles, connected by piecewise linear interpolation. **d** Cross-species agreement in chromatin state[11,28] frequency in pairs where the LECIF score is high and PhyloP score is low or vice versa. The PhyloP score is the same as in **c**. The states are the same as in Fig. 3b–d. Diagonally hatched bars show PCC from pairs with high LECIF score (>90th percentile) and low PhyloP score (<10th) in all bases within 500 bp of the human region. Horizontally hatched bars show PCC from pairs with low LECIF score (<10th) and high mean PhyloP score (>90th) in the human region. Bars are colored according to the legend on the right. Similar plots with different percentile cutoffs and also including pairs with both scores above or below the cutoffs are in Supplementary Fig. 20. Source data for **a**–**d** are provided as Source data files.Source Data

was learned using human functional genomics data with human regions that align to mouse as positive examples and the rest as negative examples ("Methods"). The human-only baseline score was weakly correlated with the human–mouse LECIF score with a PCC of 0.13 and did not reflect cross-species similarity in functional genomic features as strongly as the LECIF score (Supplementary Figs. 12, 15, and 16). These results support the contribution of mouse data to identifying conserved functional genomic properties.

**Relationship to sequence-based conservation annotations**. We next analyzed the relationship between the LECIF score and various sequence-based annotations of conservation within aligning regions. We note that while human regions that align to mouse at the sequence level do show some increase in sequence constraint relative to the entire genome, the majority of aligning regions do not show high levels of sequence constraint

(Supplementary Fig. 18). We found that human regions overlapping sequence constrained elements had a greater average LECIF score, ranging from 0.19 to 0.22 across different element sets, than the mean among human regions that align to mouse in general (0.14; Fig. 4a). When compared to five sequence constraint scores and additionally the percent identity between human and mouse, the LECIF score was moderately correlated with PCCs ranging from 0.18 to 0.25 for 50-bp windows with each score averaged across 50 bases (Fig. 4c, Supplementary Fig. 19, and "Methods"). This moderate correlation may reflect biological difference between sequence conservation and functional genomics conservation[33], although potentially also the coarse resolution and incompleteness of functional genomics data.

To provide evidence that most high LECIF scores observed in regions with low sequence constraint scores are unlikely LECIF's false positives, we analyzed human and mouse chromatin state

annotations in regions where the two scores strongly disagreed. Specifically, for pairs of regions where the LECIF score was high and the PhyloP score[8] was low in all bases within 500 bp of the human region, we computed the correlation of chromatin state frequencies as described above (Fig. 4d and Supplementary Fig. 20). We found that such pairs had strong cross-species similarity for all states, often as strong as pairs that scored high in both scores. In comparison, pairs of regions with low LECIF score and high PhyloP score had weaker cross-species similarity of frequency in all states. This suggests that the LECIF score can capture conservation at the functional genomics level even in regions that align, but have limited sequence constraint among aligning regions, potentially detecting signatures of conservation not captured by sequence constraint scores defined from multispecies sequence alignments.

To further understand the differences between the LECIF score and constraint scores, we next identified patterns within a multispecies sequence alignment that may correspond to those differences. To do this, we leveraged the ConsHMM[34] conservation state annotation of the human genome, which annotates each human genomic base based on alignment and matching patterns with vertebrate genomes in a 100-way sequence alignment (Supplementary Fig. 21). Among a hundred conservation states, the state with the highest average LECIF score corresponded to human bases that align and match to many vertebrate genomes with a moderate probability, indicating signatures of conservation across many vertebrates. This state was previously shown to most strongly enrich for promoter and CpG islands out of all conservation states. In contrast, this state had only the 12th highest average PhyloP score. This suggests that the disagreement between the LECIF score and constraint scores could be partly explained by constraint scores not capturing signatures of conservation that are actually present in the multispecies sequence alignment and further supports that the LECIF score can provide complementary information to sequence constraint scores about conservation.

Since the LECIF score prioritized the conservation state most enriched for CpG islands, which are known to have varying evolutionary dynamics at the sequence level, we analyzed the LECIF score of human CpG islands previously grouped by their distinct regimes during primate sequence evolution[35] (Fig. 4b). CpG islands in general scored high with a mean LECIF score of 0.53, and the score positively correlated with the likelihood of a CpG island being classified as slowly evolving as opposed to quickly evolving (Supplementary Fig. 22; PCC: 0.50). Slowly evolving CpG islands characterized by low rate of C-to-T deamination had higher LECIF scores with a mean of 0.65. In contrast, quickly evolving CpG islands had lower LECIF scores with a mean of 0.35. Although LECIF scores CpG islands higher than the rest of the genome in general, the score reflects the distinct evolutionary dynamics among them.

**Relationship to phenotype-associated variation**. To investigate if the LECIF score enriches for biologically important genomic loci linked to phenotype, we analyzed the relationship between the LECIF score and phenotype-associated genetic variation (Fig. 5a). We observed that regulatory disease variants from Human Gene Mutation Database (HGMD)[36] enriched for regions with high LECIF score. In contrast, we saw small depletions for common variants[37] in those high-scoring regions. We saw that high-scoring regions also exhibited enrichment of GWAS catalog[38] variants and expression quantitative trait loci (eQTLs) from GTEx[39].

We also conducted a heritability partitioning analysis with the LECIF score for 12 complex traits[40]. Specifically, we applied

heritability partitioning with an annotation of bases with a LECIF score in the top 5% in the context of a baseline set of annotations[41], which we extended to also include annotations of human regions that align to mouse and top 5% regions based on the human-only baseline score. We note that the baseline annotation set includes multiple sequence constraint annotations. We observed that the top 5% regions based on the LECIF score resulted in enrichments of heritability with statistical significance for several traits (Fig. 5b). Furthermore, we observed overall stronger enrichments for the LECIF annotation than the human-only baseline annotation and the annotation of human regions that align to mouse.

**LECIF highlights regions in mouse QTL relevant to disease**. To demonstrate how LECIF could be applied to translating biological findings, particularly in mapping trait-associated loci between mouse and human, we analyzed mouse insulin secretion QTL and human diabetes GWAS variants[42]. Previously, it was shown that human regions syntenic to the mouse insulin secretion QTL were enriched for the human diabetes GWAS variants. However, mouse QTL in general can span several megabases, making it difficult to identify likely causal variants within the loci for the trait of interest[5]. We thus mapped the mouse insulin secretion QTL to the human genome based on sequence alignment and asked whether the LECIF score could provide information in locating regions within the mapped mouse insulin secretion QTL that correspond to human diabetes GWAS variants.

We observed that human genomic windows within the mapped mouse insulin secretion QTL that overlap the human GWAS variants had a statistically higher distribution of mean LECIF scores than windows within the mouse QTL not overlapping the variants or windows overlapping the variants (Mann–Whitney $U$ test $P < 0.0001$; Fig. 6a and Supplementary Fig. 23b, c). In addition, we saw that the human diabetes GWAS variants that lie within the mapped mouse QTL had a higher distribution of mean LECIF scores than human GWAS variants outside the mouse QTL, in addition to human bases within the mouse QTL that are not the human GWAS variants (Mann–Whitney $U$ test $P < 0.0001$; Supplementary Fig. 23a). These results indicate LECIF's potential value in locating regions within mouse QTL that are more likely relevant to a given trait in human.

**LECIF highlights conserved methylation patterns linked to phenotype**. To further illustrate potential applications of LECIF, we also evaluated the ability of the LECIF score to prioritize epigenetic features conserved between human and mouse in a disease relative context. Specifically, we considered data from an epigenetic study on differential methylation in diabetic phenotypes in human and mouse[43], which was independent of the data provided to LECIF. The study identified conserved differentially methylated regions (DMRs) associated with obesity by first finding DMRs in high-fat-fed and low-fat-fed mice and then testing their homologous human regions for differential methylation between obese and lean patients. The LECIF score was significantly higher in conserved DMRs in comparison to mouse-specific DMRs (Mann–Whitney $U$ test $P < 0.01$; Fig. 6b). This supports the potential value of the LECIF score for prioritizing among all loci with epigenetic associations with phenotype in one species the specific loci whose associations are more likely to be shared in the other species.

## Discussion

We presented LECIF, a method that scores evidence for conservation between human and mouse based on a compendium of functional genomic annotations from each species. To do so,

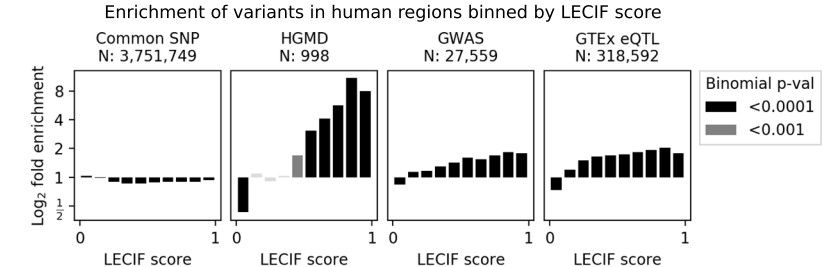

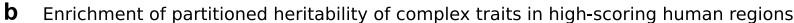

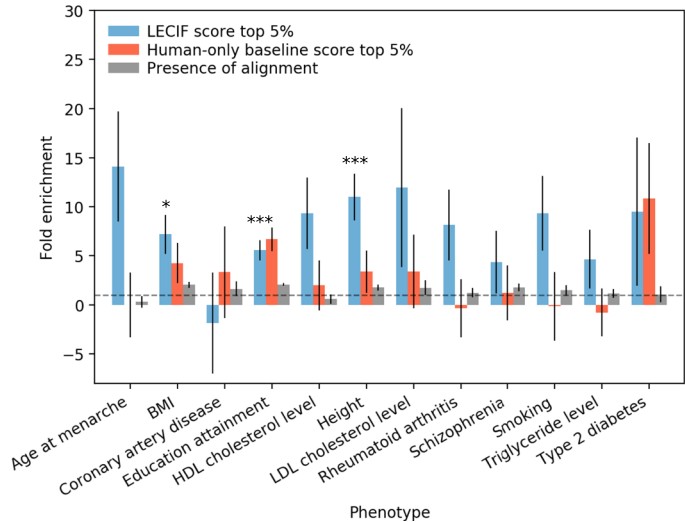

**Fig. 5 Relationship of LECIF score to genetic variants and heritability. a** Shown from left to right are plots of $\log_2$ fold enrichment for variants based on four different sets, (i) common SNPs[37], (ii) HGMD regulatory variants[36], (iii) GWAS catalog SNPs[38], and (iv) GTEx cis-eQTLs[39] across tissues, within human regions binned by the LECIF score with ten equal-width bins. Analysis was restricted to human regions that align to mouse, and a uniform background within these regions was used. Displayed above each subplot is the number of regions overlapping the variants from the corresponding set included in the analysis. Black and dark gray bars denote $\log_2$ fold enrichments that resulted in $P$ values <0.0001 and 0.001, respectively, based on one-sided bionomial tests. **b** Fold enrichments for partitioned heritability of 12 phenotypes[40] in human regions with high LECIF score. Enrichments are shown for human regions with high human–mouse LECIF score (>95th percentile; blue), and additionally for comparison regions with high human-only baseline score (>95th percentile; orange) and human regions that align to mouse (gray). Heritability partitioning[40] for the LECIF score was applied in the context of a baseline set of annotations[41], which included sequence constraint annotations and was extended to include additional annotations generated based on the human-only baseline score and sequence alignment ("Methods"). Error bars denote standard error around the enrichment estimates. Horizontal dashed line denotes no enrichment (fold enrichment of 1). * and *** denote Bonferroni-corrected one-sided $P$ values for the LECIF score annotation's enrichment <0.05 and 0.001, respectively. $P$ values and standard errors were calculated using a block jackknife over SNPs with 200 equally sized blocks of adjacent SNPs as described in ref. [40]. Source data for **a** and **b** are provided as Source data files.Source Data

LECIF trains neural networks to differentiate aligning pairs of regions from mismatched pairs of the same set of regions based on their functional genomic annotations without using sequence information as features. The functional genomic annotations include maps of open chromatin, TF binding, gene expression signals, and chromatin state annotations. The resulting score captures evidence of conservation at the functional genomics level that is based on a diverse set of annotations and thus not specific to one class of DNA elements.

We applied LECIF with >10,000 functional genomic annotations from human and mouse to learn the human–mouse LECIF score. The LECIF score had greater predictive power than several baseline scores at discriminating pairs of human and mouse regions that align to each other from mismatched pairs of aligning regions. Using H3K27ac samples matched by their tissue of origin and separately using chromatin state annotations learned jointly between human and mouse, we showed that the LECIF score reflects the relationships between biologically similar human and mouse functional genomic annotations. LECIF was able to do this without any explicit information provided about the relationship between different features within or across species. Furthermore, LECIF was able to do so even in regions where sequence constraint was low, supporting that the LECIF score provides complementary information to sequence constraint annotations. Regions with high LECIF score were enriched for phenotype-associated variants from curated databases and also for heritability of complex traits. Using matched DNA methylation samples between human and mouse and separately using matched GWAS and QTL datasets, both in the context of a diabetes trait, we showed that the LECIF score has preference for human and mouse regions with shared associations with the trait.

These results support the potential value of the LECIF score in various applications in the context of model organism research. For example, given a set of phenotypic-associated loci identified in a mouse model, which are increasingly available through efforts like the Mouse Phenome Database[44], the highest-scoring loci could be prioritized for experimental validation in human cells if possible. Conversely, given human genomic variants or candidate regulatory elements with known associations with a trait, those with the highest LECIF scores could be prioritized for

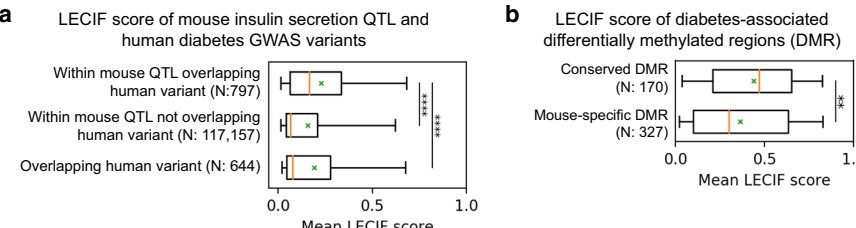

**Fig. 6 Relationship of LECIF score to genetic and epigenetic variation associated with phenotypes. a** Distribution of mean LECIF score of nonoverlapping 1-kb human genomic windows identified as lying within a mapped mouse insulin secretion QTL or containing a human diabetes GWAS variant or both[42]. "Within mouse QTL overlapping human variant" refers to windows that lie within the mouse QTL mapped to human and overlap the human diabetes GWAS variant. "Within mouse QTL not overlapping human variant" refers to windows within the mapped mouse QTL that do not overlap any human diabetes GWAS variant. "Overlapping human variant" refers to windows that overlap the human diabetes GWAS variant and lie in loci obtained by randomly permuting the locations of the mapped mouse QTL. All windows were obtained by sliding a fixed window across the QTL, and any window with less than half of its bases annotated with the LECIF score was excluded from this analysis. Displayed after each label is the number of qualified windows corresponding to that label. Each distribution is represented by a boxplot with median (orange solid line), mean (green "x"), 25th and 75th percentiles (box), and 5th and 95th percentiles (whisker). **** denotes P value <0.0001 based on a two-sided Mann–Whitney U test. Similar plots generated using different window sizes are shown in Supplementary Fig. 23. **b** Distribution of mean LECIF score in conserved differentially methylated regions (DMRs) and mouse-specific DMRs with respect to a diabetic phenotype[43]. "Conserved DMR" refers to regions with significant differential methylation (P value <0.05) in both human and mouse and the same directionality with respect to the phenotype. "Mouse-specific DMR" refers to regions with significant differential methylation in mouse, but either lacking significant differential methylation in human or showing inconsistent direction of methylation change between human and mouse. The study in which the DMRs were reported did not provide human-specific DMRs because it first identified mouse DMRs and then tested those in human and not vice versa. Displayed below each label is the number of DMRs corresponding to that label. Boxplots are formatted as in **a**. ** denotes P value <0.01 based on a two-sided Mann–Whitney U test (P = 0.003). Source data for **a** and **b** are provided as Source data files. Source Data

testing in mouse models. In addition, when loci exhibit signals of interest in both species, those with the highest LECIF scores could be prioritized for follow-up experiments.

While we expect LECIF to be useful, we do note a few limitations. LECIF only scores evidence of conservation at the functional genomics level. There thus could be regions that are conserved at the functional genomics level, but have a low LECIF score, since the evidence was not present in the data currently available to LECIF. This makes it difficult to distinguish the case of human-specific regulatory activity from insufficient evidence in the aligning mouse region's annotations based on a low LECIF score. Fortunately, the interpretation of high LECIF scores is less ambiguous. We also note that the LECIF score's resolution is limited by the resolution of the input functional genomic annotations and thus does not have the base resolution that sequence-based conservation annotations can have. In addition, LECIF is designed to aggregate information across multiple tissues and cell types and thus does not provide the direct information about a particular tissue.

In addition, we note that currently the LECIF score is only available for pairs of regions that align to each other. While in principle LECIF can be applied to score any pairs of regions, more false-positive predictions are expected as a result, compared to our presented strategy of restricting to regions that align at the sequence level. Although we explored an alternative strategy that considered non-aligning regions in a neighborhood of each pair of aligning regions, this did not lead to improvements in our evaluations over considering only the aligning regions. However, future work could develop other strategies that lead to improvements.

While here we focused on human and mouse, as mouse is a widely used model organism for human and there is substantial data available for both, LECIF can be applied to compare human to any species with a genome-wide pairwise sequence alignment to human and functional genomics data. Applying LECIF to human and mouse with mouse features downsampled demonstrated that a few hundred annotations from the nonhuman species may be sufficient to capture a large portion of conservation at the functional genomics level, although the quality of the score will depend on the coverage of the data available for the nonhuman species. As functional genomics data

from a more diverse set of species, cell types, and assays continues to become available, the utility of LECIF will continue to grow for identifying regions conserved at the functional genomics level and transferring findings from mouse and other model organism research to human biology.

## Methods

**Pairwise sequence alignment.** For the pairwise sequence alignment, we used the chained and netted alignment[6] between the human genome (hg19) and the mouse genome (mm10), with human as the reference genome for the alignment. Given multiple mouse genome segments that map to a single human genome segment, we chose the mouse segment with the highest alignment score. This alignment was obtained from the UCSC Genome Browser[45].

**Functional genomics data used for input features.** ChromHMM[28] chromatin state annotations for human were from the 25-state model learned for 127 cell and tissue types based on imputed data from the Roadmap Epigenomics Project[14], and for mouse from the 15-state model learned for 66 cell and tissue types from ENCODE[46]. Peak calls for DNase-seq and ChIP-seq experiments of TFs, histone modifications, and histone variants were from Roadmap Epigenomics[14], ENCODE[13], and Mouse ENCODE[11]. Peak calls for CAGE experiments were from FANTOM5[29]. RNA-seq signal data were from ENCODE[13] and Mouse ENCODE[11]. For ENCODE and Mouse ENCODE data, we used the uniformed processed version available from the ENCODE portal. Additional information including the specific source of each dataset used is listed in Supplementary Data 1.

**Defining pairs of human and mouse regions for training and prediction.** To define pairs of human and mouse regions for training and prediction for LECIF, we first identified alignment blocks from the pairwise alignment. We defined alignment blocks as pairs of human and mouse genomic segments without any alignment gap, meaning the human and mouse genomic segments both had a nucleotide present at each base in the block. We then for each alignment block defined nonoverlapping windows of 50 bp starting from the first base in the alignment block. Each 50-bp window defined a region. If the alignment block ended within the 50-bp window, we truncated the window to the end of the block to define the region. This resulted in some regions being shorter than 50 bp. To define negative examples, we randomly paired up human and mouse regions included in the positive examples. With this procedure, all human regions included in the negative examples aligned somewhere else in the mouse genome, and all mouse regions in the negative examples aligned somewhere else in the human genome.

**Defining subsets of pairs of regions for training and evaluation.** All human and mouse chromosomes, except for Y and mitochondrial chromosomes, were used. X chromosomes were excluded from training, validation, and test, but included for

prediction and downstream analyses. To generate predictions for all pairs of human and mouse regions that included a human region from an even chromosome or X chromosome, we trained LECIF on pairs of human and mouse regions, such that both the human and mouse regions came from a subset of odd chromosomes for its respective species (Supplementary Data 2). To form a validation set, which we used for hyper-parameter tuning and early stopping during training, we used pairs of regions such that the human region came from a subset of odd chromosomes not used in training and likewise for mouse. To form a test set, which we used to generate the receiver operating characteristic (ROC) and precision-recall (PR) curves, we used all pairs of regions such that both the human and mouse region were from an even chromosome. To generate predictions for all pairs that included a human region from an odd chromosome, we took an analogous approach as above (Supplementary Data 2). There was no overlap in genomic regions used for training, validation, and test. To assess the agreement between a model trained on odd chromosomes and a model trained on even chromosomes, we used pairs of regions that were from a subset of chromosomes not used in training or validation of either model (Supplementary Data 2).

**Feature representations**. For each pair of human and mouse regions, we generated two feature vectors. The two vectors were based on annotations overlapping the first base of the human and mouse regions, respectively, which were at most 50 bp. For computational considerations, we only used the first base of each region to provide the LECIF score for all bases in the region. To evaluate the effect of this, we computed the PCC between a score defined at base resolution for 1 million randomly sampled pairs of human and mouse bases that align to each other and the LECIF score, which was defined at every 50 bp within each alignment block, for the same set of 1 million pairs.

Each peak call corresponded to one binary feature. If a base overlapped a peak call for an experiment, the corresponding value in the feature vector was encoded as a 1, otherwise it was encoded as a 0. While real-valued signals are also available for these experiments with peak calls, we used the binary peak calls for improved scalability and reduced potential for overfitting. Chromatin state annotations were one-hot encoded such that there was a separate binary feature, representing the presence of each chromatin state in each cell or tissue type. Each RNA-seq experiment corresponded to one continuous feature. For human RNA-seq experiments, to also have the features in the range 0–1, we first computed the maximum and minimum signal value at any base in any of the human RNA-seq experiments. We then normalized values by subtracting the minimum signal value and dividing by the difference between the maximum and minimum signal values. We separately did the same normalization for mouse RNA-seq experiments. In total, we used 8824 human features and 3113 mouse features. Number of features from each data type are reported in Supplementary Data 1.

**LECIF classifier**. The classifier that LECIF uses is an ensemble of neural networks where each neural network had a pseudo-Siamese architecture[47] (Supplementary Fig. 24). A Siamese neural network consists of two identical subnetworks followed by a final subnetwork that combines the output from the two subnetworks to generate a final prediction[48]. A pseudo-Siamese network is similar except it uses two distinct subnetworks instead of identical subnetworks. In LECIF, the two subnetworks corresponded to human and mouse. Human and mouse feature vectors were given to the human and mouse subnetworks, respectively, as input. We also evaluated using a fully connected neural network, but found that it led to highly similar predictions (PCC: 0.95), while taking longer to train.

Hyper-parameters of a neural network consisted of number of layers in each subnetwork and the final subnetwork, number of neurons in each layer, batch size, learning rate, and dropout rate. To set the values of the hyper-parameters, we conducted a random search, where we generated 100 neural networks, each with different randomly selected combinations of hyper-parameters (Supplementary Data 3). Each neural network was trained on the same set of randomly selected 1 million positive and 1 million negative training examples. We applied 50 times more weight to our negative examples than positive examples during training so that a high LECIF score corresponds to strong evidence of conservation. We identified the best-performing combination of hyper-parameters based on maximizing the AUROC on the validation examples.

With the best-performing combination of hyper-parameters, we then trained a new set of 100 neural networks, each provided with different subsets of 1 million positive and 1 million negative training examples randomly selected from a pool of all training examples (>2.2 million positive and >2.2 million negative). While the same genomic regions in each species appear in both positive and negative examples given all available training examples, a single neural network may not necessarily encounter the same set of regions in its positive and negative examples due to random sampling. We applied the same increased weighting of negative examples as above. The final prediction of the ensemble was the average of the predictions from the 100 trained neural networks.

For both hyper-parameter search and training, we stopped training if there were no improvements in AUROC evaluated on the validation examples over three epochs. We saved the classifier from the epoch with the highest AUROC on the validation examples. The maximum number of epochs we allowed during training was 100, and the maximum training time we allowed was 24 h.

We also generated a version of the LECIF classifier, LECIF-GB, which was trained in the same way as LECIF except the negative examples were pairs of human and mouse regions that were both randomly selected from anywhere in their respective genomes, as opposed to being constrained to aligning regions. We used PyTorch (version 0.3.0.post4)[49] for implementation of the neural networks.

**Random forest baseline**. We trained, applied, and evaluated RF using the same procedure as explained above, except we used a decision tree in place of a neural network. We also did hyper-parameter selection as explained above, but for a set of hyper-parameters unique to decision trees (Supplementary Data 3). We used Scikit-learn (version 0.19.1)[50] for implementation.

**Canonical correlation analysis baseline**. We trained an ensemble of CCA mappings using the same procedure as above, except using a CCA mapping in place of a neural network and positive examples only. We applied and evaluated the ensemble using the same procedure as explained above. We also did hyper-parameter selection as explained above, but for a set of hyper-parameters unique to CCA mapping (Supplementary Data 3) and through a grid search instead of random search. We used Pyrcca[51] for implementation. Similarly, we also trained an ensemble of DCCA mappings[52]. We did hyper-parameter selection as done for CCA, but for a set of hyper-parameters unique to DCCA mapping and through a random search (Supplementary Data 3). We used a MATLAB implementation of DCCA from prior work[53].

**Logistic regression baseline**. We trained, applied, and evaluated an ensemble of LR classifiers using the same procedure as above, except we used a LR classifier in place of a neural network. We also did hyper-parameter selection as for the neural networks, but for a set of hyper-parameters unique to LR models (Supplementary Data 3) and through a grid search instead of random search. We used Scikit-learn (version 0.19.1)[50] for implementation.

**LECIF scores separately learned for coding and noncoding bases**. We trained, applied, and evaluated two models using separate training data from coding and noncoding bases. Training examples used to learn the original LECIF score were grouped into coding and noncoding examples based on whether each example's human region overlapped GENCODE annotation of coding sequence. Given noncoding training examples, the same learning procedure used to learn the original LECIF score was used to learn a score from noncoding regions. For coding training examples, all available training and validation examples (~40,000 training and ~20,000 validation examples) were used for hyper-parameter search. Given optimized parameters, each classifier was trained on 10,000 positive and 10,000 negative training examples, instead of 1 million for each. These adjustments were made specifically for training a model on coding regions because there were much fewer regions to use.

**LECIF scores with fewer mouse features**. We trained, applied, and evaluated two models that used the same human features as LECIF, but used fewer mouse features. One of the models used 10% of the original set of mouse features and the other used 1%. Except for downsampling features, model training, and hyper-parameter search were done the same way as LECIF with the full set of features. To select mouse features for the 10% model, we first randomly selected 6 out of 66 epigenomes in the 15-state mouse ChromHMM chromatin state annotations, resulting in 90 one-hot encoded features corresponding to chromatin states. We then randomly sampled 221 features from features corresponding to mouse DNase-seq, ChIP-seq, RNA-seq, and CAGE annotations, resulting in 331 mouse features in total. For the 1% model, we randomly sampled 31 features from those corresponding to mouse DNase-seq, ChIP-seq, RNA-seq, and CAGE annotations. We did not use any features corresponding to chromatin state annotations in the 1% model. This allowed us to simulate LECIF's application to a nonhuman species with limited functional genomic data, where chromatin state annotations are not available. As in training, only the selected mouse features along with the full set of human features were used for prediction and evaluation for these scores based on fewer mouse features.

**Human-only baseline**. We trained, applied, and evaluated a human-only baseline, which used the same human features as LECIF, but did not use any mouse features and used a different set of positive and negative examples for training. The positive examples were human regions that align to the mouse genome and the negative examples were human regions that do not align to the mouse genome. We otherwise used the same procedure for training, prediction, and evaluation as for LECIF except we used an ensemble of fully connected neural networks. We also did hyper-parameter selection as for LECIF, but for a set of hyper-parameters of a fully connected neural network (Supplementary Data 3). We used PyTorch (version 0.3.0.post4)[49] for implementation.

**Area under the ROC and PR curves**. To compute each classifier's classification performance based on area under the ROC curve and PR curve, we used Scikit-learn's implementation[50].

**Defining LECIF score including adjacent non-aligning mouse regions**. To generate a LECIF score for each pair of a human region and its aligning mouse region with adjacent non-aligning mouse regions also considered, we computed LECIF scores for additional pairs that consisted of the same human region and distinct 50-bp mouse regions located within a neighborhood of $W$ bases centered around the aligning mouse region (Supplementary Fig. 3). The non-aligning mouse regions were defined by sliding a 50-bp window from the first base of the aligning mouse region in both the 5′ and 3′ directions. We then took the maximum over these LECIF scores to produce a score which we refer to as the region-neighborhood LECIF score. We varied $W$ between 0 and 20 kb. We note that $W$ of 0 corresponds to the original LECIF score.

**Computing mean LECIF score for chromatin states**. To compute the mean LECIF score for each chromatin state in the 25-state ChromHMM annotation across 127 human epigenomes[14,28], for every pair of chromatin state and epigenome, we first averaged the LECIF score in all aligning regions annotated by the state in the epigenome. We then for each chromatin state computed the average of 127 mean scores, each coming from an epigenome.

**H3K27ac activity similarity**. To define the H3K27ac activity similarity between human and mouse based on known biology, we took all human and mouse H3K27ac experiments used for features and manually grouped them into the following 14 tissue type groups based on available annotations of the experiments: adipose, bone element, brain, embryo, heart, intestine, kidney, limb, liver, lung, lymph node, spleen, stomach, and thymus. Supplementary Data 1 specifies which experiment was assigned to which group, but we note that information about these groups were not used in learning the LECIF score. The 14 groups listed above were represented in at least one H3K27ac experiment in both species. For the analysis, we discarded experiments that did not belong to any of the tissue groups.

For each pair of human and mouse regions, we then defined vectors $h$ and $m$ of length 14 where $h_i$ and $m_i$ correspond to the fraction of experiments in the $i$th group with peak calls that overlapped the human and mouse regions, respectively. Finally, for each pair of human and mouse regions, we computed the weighted Jaccard similarity coefficient[54] between these two vectors. The weighted Jaccard similarity coefficient is defined as:

$$J(h, m) = \frac{\sum_i \min(h_i, m_i)}{\sum_i \max(h_i, m_i)} \quad (1)$$

Any pair with an undefined similarity coefficient due to the denominator summing up to zero was removed from the analysis.

**Chromatin state frequency correlation**. To analyze cross-species agreement of chromatin state frequencies as a function of the LECIF score, we first grouped pairs of human and mouse regions based on their LECIF score. When binning based on either score, five or ten equal-width bins were used with varying numbers of pairs in each bin. We repeated the procedure when using the human-only baseline score in place of the LECIF score. We also binned based on the percentile rank of scores, where either five or ten bins were used with nearly the same number of pairs in each bin.

To compute the chromatin state frequency correlation across a set of pairs of human and mouse regions defined as described above, we used a chromatin state model jointly learned from both human and mouse genomes[11]. For each of the seven chromatin states, we defined vectors for human and mouse. An element of a vector for human corresponds to the fraction of epigenomes, in which one of the human regions is annotated with the state, and similarly for the mouse vector and regions. We then computed the PCC between the two vectors for each chromatin state, resulting in seven PCC values.

**Correlation between the LECIF score and sequence constraint scores**. To compute the correlation between the LECIF score and sequence constraint scores, we slid a 50-bp genomic window in 10-bp increment across the human genome. For each window, we computed the mean of each score (LECIF or sequence constraint). For each sequence constraint score, we computed the PCC and SCC between the LECIF score and the sequence constraint score for windows with at least $n$ bases annotated by the two scores, with $n$ ranging from 1 to 50. The two scores were not required to be defined on the same set of bases within the 50-bp window.

**Heritability partitioning analysis**. To perform the heritability partitioning analysis, we used the LD-score regression software ldsc (v1.0.0)[40]. We generated an annotation of all human regions that align to the mouse genome and have a LECIF score above the 95th percentile. We used this annotation in the context of the baseline annotation set (v2.1) from Gazal et al.[41] along with another annotation generated based on the human-only baseline score instead of the LECIF score as well as an annotation of human regions that align to the mouse genome. We also included 500-bp windows around each annotation to dampen the inflation of

heritability in neighboring regions due to linkage disequilibrium, following the procedure in ref. [40].

We applied ldsc to this extended set of 60 annotations for the following 12 traits[40]: age at menarche, body mass index, coronary artery disease, education attainment, HDL cholesterol level, height, LDL cholesterol level, rheumatoid arthritis, schizophrenia, smoking, triglyceride level, and type 2 diabetes.

**Reporting summary**. Further information on research design is available in the Nature Research Reporting Summary linked to this article.

## Data availability

The human–mouse LECIF score is available at https://github.com/ernstlab/LECIF. Links to data files used to generate input features to LECIF are listed in Supplementary Data 1. The human–mouse pairwise alignment is available at http://hgdownload.cse.ucsc.edu/goldenpath/hg19/vsMm10/axtNet/. For TSS, gene body, intron, exon, coding exon, 5′ UTR, and 3′ UTR annotations, we used GENCODE annotations V31lift37 for human and VM23 for mouse. We downloaded these annotations along with classification of evolutionary dynamics of CpG islands[35] and common SNPs (dbSNP v7)[37] from the UCSC Table Browser[45]. The HGMD variants that we used were variants annotated as "regulatory mutations" in the April 2012 public release of HGMD database[36,55]. The following URLs contain datasets that were used in the heritability partitioning analysis: baseline annotation set[41]: https://storage.googleapis.com/broad-alkesgroup-public/LDSCORE/1000G_Phase3_baselineLD_v2.1_ldscores.tgz; age at menarche[56]: https://www.reprogen.org; body mass index and height[57]: http://www.broadinstitute.org/collaboration/giant/index.php/GIANT_consortium_data_files; coronary artery disease[58]: http://www.cardiogramplusc4d.org/data-downloads; education attainment[59]: https://www.thessgac.org/data; HDL cholesterol level, LDL cholesterol level, and triglyceride level[60]: http://csg.sph.umich.edu/willer/public/lipids2010; rheumatoid arthritis[61]: http://plaza.umin.ac.jp/yokada/datasource/software.htm; schizophrenia[62] and smoking[63]: www.med.unc.edu/pgc/downloads; and type 2 diabetes[64]: http://www.diagram-consortium.org/downloads.html. Source data are provided with this paper.

## Code availability

The LECIF software is available at https://github.com/ernstlab/LECIF.

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

## Acknowledgements

We thank the ENCODE, Mouse ENCODE, Roadmap Epigenomics, and FANTOM consortia for generating data and making it publicly available. We thank members of the Ernst lab for useful discussions. We acknowledge funding from US National Institutes of Health (DP1DA044371 and U01MH105578 to J.E.); US National Science Foundation (CAREER Award #1254200 to J.E.); Kure It Cancer Research (Kure-IT award to J.E.), and a Rose Hills Innovator Award (J.E.).

## Author contributions

S.K. and J.E. developed the method, analyzed the results, and wrote the paper.

## Competing interests

The authors declare no competing interests.
