## [Peer Review File · Nature Communications]

Reviewers' Comments:

Reviewer #1:

Remarks to the Author:

Kwon and Ernst present LECIF, a computational method that identifies genomic regions with similar functional activity between human and mouse, as measured by 100s of sequencing-based genomics assays in each species. A simple solution to this problem would be to find a matching set of experiments and evaluate the similarity of the experiments for each locus. However, such matching is difficult.

LECIF gets around this issue by training a classification model to distinguish pairs of mapping regions from random pairs.

The work is impactful and timely. The manuscript includes many computational experiments, which clarify the strengths and limitations of LECIF. I have several comments and questions, none of which is major.

1. The authors state "we expect LECIF to be useful", but there is little direct explanation of what specific downstream tasks it solves. Many of the results hint at potential applications, but, as far as I can tell, the only one that mentions a realistic task is section "LECIF score highlights regions within mouse quantitative trait loci relevant to human disease". I think such tasks exist, but they should be spelled out explicitly.

2. The LECIF model uses an average of 100 neural networks, each trained on a different subset of training data. The justification for using 100 neural networks is unclear. While the authors did demonstrate that 100 neural networks performed better than 1 neural network, there was little investigation to determine the optimal number of neural networks or the tradeoff between size of network and number of separate networks. Analysis of the distribution of the output of these networks would be useful in understanding how they contribute to the LECIF score, and why they performed better than 1 neural network.

3. The model uses DNase-seq, ChIP-seq, and CAGE experiments as binary features defined by peak calls. Why were binarized values used instead of (or in addition to) real-valued signals?

4. An alternative approach would be to train a model to predict human data from mouse (using mapping regions), then evaluate functional similarity as the similarity between observed and predicted. I would be curious how this compares. A related approach is Deep Canonical Correlation Analysis (Deep CCA), the neural network extension of CCA, the latter of which the authors evaluated as a baseline. Note that none of these have been applied to the authors' problem before.
<http://proceedings.mlr.press/v28/andrew13.html>

5. What is the dependence of LECIF accuracy on the number of input data sets? This is important for its utility in application to species with less available data than mouse.

Minor points:

- This sentence is difficult to follow due to pronouns in "other species" and "each other": "For training data, positive examples are pairs of regions between the two species that align at the sequence level while negative examples are randomly mismatched pairs of human and mouse regions that align somewhere in the other species, but not to each other." It was initially unclear what was meant due to the excessive use of pronouns.

--

Maxwell Libbrecht, School of Computing Science, Simon Fraser University

Subreviewer: Mariam Arab, School of Computing Science, Simon Fraser University

Reviewer #2:

Remarks to the Author:

Kwon and Ernst developed an interesting machine-learning model to quantitatively define the functional conservation of sequences, between mouse and human. The approach is very solid as they integrated over 1000 datasets, including ChIP-Seq for histone modifications and various TF binding profile. The machine part is well written and detailed. This group has a well-established reputation of develop and release the software to the public (including one of the best known epigenomic software, ChromHMM). Technically I do not have much to add.

There is no doubt that the LECIF is important. But what I hope the authors to clarify is: how are we going to use the LECIF score in interpreting biological discoveries or use it in experiment design? Does it help us identify tissue-specific or species-specific cis-regulatory elements?

Below are some minor technical questions:

1. Can authors split the training site into non-coding and coding? I want to understand where the lower LECIF score genes come from.
2. Page 10 "... In contrast, low-scoring pairs of regions were annotated with dissimilar sets of states in human and mouse and more frequently annotated with the quiescent state than high-scoring pairs ". On the other hand, can authors plot the distribution of score for the unmarked region and all the chromatin states? This can strengthen their claim.
3. It would be interesting to investigate the LECIF scores across TADs boundaries.

Reviewer #3:

Remarks to the Author:

Review of:

Learning a genome-wide score of human-mouse conservation at the functional genomics level

by Soo Bin Kwon & Jason Ernst

=====
=====

The authors describe a computational approach for assessing the conservation between human and mouse genomic regions, in terms of genome-wide functional genomics datasets.

In particular, they present a largely data-driven method that, very importantly, does not rely on an explicit mapping between, or even matching of, underlying datasets between species.

This is a conceptual advance over existing approaches, and particularly relevant for current large-scale data generation efforts (e.g. ENCODE), which tend to be sub-optimally coordinated in terms of matching assays between species.

Overall, the manuscript is clearly written, albeit a bit dry.

I feel that the writing and messaging of the paper can be improved by making the goals of each described analysis, and the problems they address, much more explicit.

As it stands, the manuscript reads like most of the "meat" is front-loaded, with more technical (control) analyses in the second half of the paper.

Perhaps the manuscript can be shortened in a way that puts full focus on the main message, namely the fully data-driven approach for dealing with the important question of (functional) conservation between species.

The core statistical and machine learning approaches used are acceptable, and performance evaluations are generally executed according to the latest best-practices.

That said, there are a number of concerns I have with some of the analyses and the conclusions the authors reach based on these, highlighted in the following points below.

Points:

1.

Page 2, start of first paragraph.

It was not clear to me what the exact distinction is here between "pairwise sequence alignment" (regional conservation, allowing for gaps?)

and "constrained at the sequence level" (per-basepair conservation?)

My understanding is that the current work builds upon the former, but since the method is very much anchored on

the presence of some degree of sequence conservation, it is important to make this very clear at this point in the manuscript.

Also, perhaps I missed this, but what is the starting basis for the fraction of the human genome that can be used with LECIF in this way? 40%?

2.

It should be made more clear at an early point in the manuscript what exactly the working resolution of LECIF in this study is (50bp?).

Although in the manuscript (page 4), it is mentioned that the /predictions/ are done at every 50bp, it is not made clear (enough) whether this is also the case for the training phase.

3.

Page 4, Two million positive and negative examples.

Does this mean that the same (two million) genomic regions in each species were re-used for positive (aligning) and negative (non-aligning) sets?

And if so, assuming a resolution of 50bp, this training set encompasses 100Mb of the human and mouse genomes?

4.

Page 7, first paragraph.

The authors mention that although LECIF can be used on genomic regions with no sequence conservation between species,

doing so would result in a higher false positive rate. I would like the authors to elaborate on this in more detail.

In particular, is this because non-conserved regions are less likely to be functional to begin with, or is there a more structural (e.g. sequence composition?) difference between (sequence-based) conserved and non-conserved regions?

After all, later the authors mention that LECIF scores are often high when sequence conservation is low ("lacking", see point below).

5.

Perhaps I missed this in the Methods, but how did the authors deal with unambiguous conservation mappings between human and mouse, e.g. multi-mapping regions?

6.

Figure 1b suggests that predictions are only done on aligning pairs.

During model building and evaluation of course also non-aligning pairs need to be considered.

Can the authors explain their reasoning for ultimately applying the model only to aligning pairs more clearly?

Perhaps one thought related to this is what would happen if for instance the human-only baseline model

(although clearly not performing as well as the full model) would be run genome-wide.

7.

Weighting of positive and negative examples during classification: why 50x more weight on negative examples?

I appreciate the effort the authors have made to compare this scenario to one where positive and negative examples are weighted equally (Supplementary Figure 1),

but it remains unclear to me exactly why the 50x weighting is preferable over the alternative.

Additionally, this weighting has a profound influence on the LECIF score distribution (see same figure), which begs the question: how was the 50x determined?

Why not 10x, 100x?

8.

Page 7, first paragraph.

The case of regulatory element turnover, i.e. lacking strict (positional) sequence conservation, is an important one.

However, I am not convinced that the local translation approach proposed sufficiently takes care of that concern, due to two reasons.

One, the 50bp size and non-overlapping nature of the considered neighboring windows are arbitrary with respect to possible turnover events,

and are therefore expected to "capture" any potential turnover events with only a small probability.

Two, the 50bp resolution allows for plenty of more complex turnover events,

not restricted to what is currently considered essentially a linear full-window translation event.

I can't help but wonder whether there would be a better way to do this, e.g. by comparing the region-region score to a region-altregion score

where altregion is a region (or regions) that contains a proportional increase of sequence conservation at a single-basepair level.

9.

Page 8, distribution of LECIF scores over input features

As it is, this section does not provide much new insight in how the LECIF score behaves.

It seems an obvious miss to not mention the clear dichotomy between features associated with (well-conserved) elements like promoters (CAGE, H3K4me3, TSS-states) and (less conserved)

elements like enhancers (DNase-seq, ChIP, H3K27ac, enhancer-states).

This would drive the point home that LECIF is (on average) pointing towards functionally well conserved elements.

All-in-all, I would strongly consider including this as a single sentence in the manuscript, instead of two full paragraphs and a figure.

10.

The match-analyses described on Page 9 & 10 are more informative, as they are core to the message of the paper that explicit matching of datasets between species may not be necessary.

Figure 4 nicely conveys this point, however for panels c & d I would be curious to see what the >95% and <5% LECIF score scenarios would look like for non-aligning pairs.

An important point in this section, that is in my opinion almost glanced over, is the suggestion that having only human functional genomics data is not sufficient.

Perhaps the evidence for this (mostly Supplementary Figure 10) is not strong enough to warrant a

stronger statement on this,
but perhaps this analysis can be tightened a bit and this conclusion can be drawn more explicitly.

11.

The question of whether sequence conservation correlates with "functional genomics" conservation is an interesting and important one.

The authors address this, in part, by studying the LECIF score distribution relative to overlap with conserved elements and bases called by various computational methods (Figure 5).

However, I think the leftover ambiguity mentioned in point 1 above is not helping the authors to make their point.

The (combinations of) metrics used (i.e., overlapping elements, conservation methods score strength, number of conserved bases, etcetera)

coupled with the weak observed effects (i.e. PCC/SCC, mean/median differences), leaves the reader with a lot of doubt and questions.

Can the authors come to a more solid statement based on these results?

12.

The second part of this section provides a partial solution by explicitly considering regions that are discordant between sequence conservation and LECIF scores.

It is encouraging to see that regions with low sequence conservation (according to phyloP) but high LECIF scores tend to show good inter-species chromatin state agreement.

Yet, the authors then mention:

"This suggests that the LECIF score can capture conservation at the functional genomics level even in regions that lack sequence constraint,

potentially detecting signatures of conservation not captured by sequence constraint annotations."

This provides a bit of a conundrum given how LECIF scores are only calculated for regions that already align

(i.e. have some level of sequence conservation) between human and mouse.

Perhaps the word "lack" is too strong here? Again, this comes back to point 1 above, and clarity of definitions used.

13.

The authors go on to show association of LECIF score with disease-relevant genetic variation.

This was done by performing LD score regression with partitioned heritability (S-LDSC), where one of the included annotations consisted of high-scoring LECIF regions.

While the presented enrichment levels are convincing, this does not show how much high-scoring LECIF regions contribute to heritability, controlling for all other annotations.

The discussion mentions conditioning on (e.g.) sequence constraint annotation, but the heritability enrichment levels of S-LDSC do not necessarily reflect this.

Instead, looking at the coefficients (tau values) of the model to get a clearer view of the contribution specific to LECIF would have been worthwhile.

Reviewer #1 (Remarks to the Author):

Kwon and Ernst present LECIF, a computational method that identifies genomic regions with similar functional activity between human and mouse, as measured by 100s of sequencing-based genomics assays in each species. A simple solution to this problem would be to find a matching set of experiments and evaluate the similarity of the experiments for each locus. However, such matching is difficult. LECIF gets around this issue by training a classification model to distinguish pairs of mapping regions from random pairs.

The work is impactful and timely. The manuscript includes many computational experiments, which clarify the strengths and limitations of LECIF. I have several comments and questions, none of which is major.

We thank the reviewer for the summary and the positive comments. We also thank the reviewer for the constructive comments and questions that have improved the paper. We describe below how we addressed them.

1. The authors state "we expect LECIF to be useful", but there is little direct explanation of what specific downstream tasks it solves. Many of the results hint at potential applications, but, as far as I can tell, the only one that mentions a realistic task is section "LECIF score highlights regions within mouse quantitative trait loci relevant to human disease". I think such tasks exist, but they should be spelled out explicitly.

We thank the reviewer for raising this point and apologize that this was not clear. We have edited two sections in the Results as well as the Discussion to more explicitly spell out how LECIF can be used in various tasks as follows.

First, we edited the last sentence of section 'LECIF score highlights regions within mouse quantitative trait loci relevant to human disease' to read:

"These results indicate LECIF's potential value in locating regions within mouse QTL that are more likely relevant to a given trait in human."

In addition to the section on QTL that the reviewer mentioned, the following section 'LECIF score highlights regions with conserved differential methylation patterns linked to phenotype' also demonstrates how the score could be useful, for example when prioritizing phenotype-associated human loci to test in a mouse model. We apologize that this was not made clear. To explicitly state the purpose of this section's analysis, we have edited the first sentence of this section to read:

"To further illustrate applications of LECIF, we also evaluated the ability of the LECIF score to prioritize epigenetic features conserved between human and mouse in a disease relative context."

Lastly, we added the following sentences to the Discussion to be more explicit on how the LECIF score overall could be applied to realistic tasks:

“These results support the potential value of the LECIF score in various applications in the context of model organism research. For example, given a set of phenotypic-associated loci identified in a mouse model, which are increasingly available through efforts like the Mouse Phenome Database, the highest-scoring loci could be prioritized for experimental validation in human cells if possible. Conversely, given human genomic variants or candidate regulatory elements with known associations with a trait, those with the highest LECIF scores could be prioritized for testing in mouse models. In addition, when loci exhibit signals of interest in both species, those with the highest LECIF scores could be prioritized for follow-up experiments.”

2. The LECIF model uses an average of 100 neural networks, each trained on a different subset of training data. The justification for using 100 neural networks is unclear. While the authors did demonstrate that 100 neural networks performed better than 1 neural network, there was little investigation to determine the optimal number of neural networks or the tradeoff between size of network and number of separate networks. Analysis of the distribution of the output of these networks would be useful in understanding how they contribute to the LECIF score, and why they performed better than 1 neural network.

We thank the reviewer for the suggestion. We have now added **Supplementary Figure 3** which shows the effect of the number neural networks on LECIF’s classification performance and prediction robustness, as shown below.

Supplementary Figure 3. Effect of the number of ensembled neural networks on predictive power and robustness.

Analysis of the effect of the number of neural networks used in LECIF, which trains an ensemble of 100 neural networks (NN), on classification performance and robustness of predictions. LECIF's predictive performance and robustness are compared to those of ensembles with fewer NN.

a. Effect of the number of NN in an ensemble on the area under the receiver operating characteristic curve (AUROC). Given 100 individual NN trained in LECIF, for each number of NN shown in the x-axis, x , we select at most 100 different ensembles, each of which is a combination of x NN. If there are 100 or fewer possible combinations, all are used. Otherwise, 100 combinations are randomly selected from all possible combinations. For each ensemble, we generated its prediction for test data by averaging the predictions from its NN. This test data was held out from training and validation of the NN. We finally computed AUROC for each ensemble and obtain the mean AUROC for each x by averaging the AUROCs over all ensembles consisting of x neural networks. Negative examples were weighted 50 times more than positive examples when computing AUROC.

b. Similar to **a** except showing area under the precision-recall curve (AUPRC) instead of AUROC. The same procedure and test data were used as **a**.

c. Similar to **a** except showing PCC between scores learned from different training data instead of AUROC. Ensembles were selected as done in **a** except we generated their predictions for held-out data that was excluded from all training, validation, and test (**Methods**). Given the ensembles generated for each number of NN shown in the x-axis, we computed PCC between scores predicted by two ensembles, each trained on non-overlapping training data. The two ensembles are matched randomly if there are multiple ensembles with the same number of NN but trained on different data. We then computed the mean PCC for each number of NN by averaging the PCCs over the pairs of ensembles.

d. Similar to **c** except showing SCC instead of PCC.

We observed that both classification performance and prediction robustness increase with more neural networks. However, there is diminishing return with increasing the number of neural networks, but fixed increases in computational cost and time. Based on our analysis, using more than 100 neural networks would provide minor gains. We note that to further optimize computational cost it would be possible to reduce the number of networks to 10 with only slightly worse performance. In the revised manuscript, we now note this in section 'Comparative evaluation of LECIF's predictive performance' as follows:

"We used LECIF with an ensemble of 100 neural networks and confirmed it led to better performance than using fewer networks, although fewer networks could be used to save computational cost with a small decrease in performance (**Supplementary Figs. 2-3**)."

We note that the tradeoff between network size and number of networks was not explored in our work because we ran hyperparameter search on a single neural network and then trained an ensemble of optimized neural networks. Running a hyperparameter search on an ensemble of neural networks instead where the number of networks is included as a hyperparameter along with network size would have taken significantly more computational resources so we did not pursue this.

3. The model uses DNase-seq, ChIP-seq, and CAGE experiments as binary features defined by peak calls. Why were binarized values used instead of (or in addition to) real-valued signals?

We thank the reviewer for the question. The use of binary features has the advantage that the method is more scalable since it is substantially less computationally

demanding to store and retrieve binary data than real-valued data. Given the large amounts of data available in both human and mouse and a fixed amount of computational resource, we expect more gains in incorporating more binary datasets than using fewer real-valued datasets, particularly for assays that are often analyzed as peaks, such as DNase-seq, ChIP-seq and CAGE data. Moreover, when an input has a binary value the classifier is less likely to overfit signal variation that may correspond to noise. We note that LECIF software can handle continuous values, which we do for RNA-seq experiments since it is more common to use quantitative values than peaks calls from this assay. We now mention this in the following sentence in section 'Feature representations' in Methods:

“While real-valued signals are also available for these experiments with peak-calls, we used the binary peak calls for improved scalability and reduced potential for overfitting.”

4. An alternative approach would be to train a model to predict human data from mouse (using mapping regions), then evaluate functional similarity as the similarity between observed and predicted. I would be curious how this compares. A related approach is Deep Canonical Correlation Analysis (Deep CCA), the neural network extension of CCA, the latter of which the authors evaluated as a baseline. Note that none of these have been applied to the authors' problem before.

<http://proceedings.mlr.press/v28/andrew13.html>

We thank the reviewer for the suggestions. We now compare to Deep CCA (DCCA) as an additional alternative method in **Figure 2c,d**, which we also show below. DCCA performs slightly better than CCA, but not better than using random forest or LECIF.

Figure 2. Characteristics of the LECIF score.

c. Receiver operating characteristic (ROC) curve comparing LECIF, random forest (RF), canonical correlation analysis (CCA), deep canonical correlation analysis (DCCA), and logistic regression (LR) for classifying pairs of regions that align at the sequence level, evaluated on a common set of held-out test data. Shown on the $y=x$ line is random expectation. Legend indicates color and mean area under the ROC curve (AUROC) corresponding to each method. The ROC curve and AUROC of each method was obtained by classifying 100,000 positive and 100,000 negative examples randomly sampled with replacement from all available test examples 100 times. Negative examples were weighted 50 times more than positive examples. For each method, standard deviation of the 100 AUROC values was under 0.005.

d. Similar analysis as in **c** except for precision-recall (PR) curves instead of ROC curves. Standard deviation of the 100 area under the PR curve (AUPRC) values was also under 0.005 for all methods.

The other suggestion of predicting individual human datasets from mouse is intriguing, but we feel it primarily addresses a different problem than the one we address in this paper and thus best left to future work. In this paper, we focus on the problem of learning an overall evidence of functional genomics conservation score between human and mouse. Predicting individual human datasets would create a lot of computational overhead if one is just interested in an overall score. Additionally, it is unclear how an overall score would be best defined from the individual predicted human datasets.

5. What is the dependence of LECIF accuracy on the number of input data sets? This is important for its utility in application to species with less available data than mouse.

We thank the reviewer for the question. We now show a comparison between the LECIF score learned with all features and two alternative scores learned with the same human functional genomic features and fewer mouse functional genomic features in **Supplementary Fig. 5** shown below. With only 311 mouse features instead of 3,113, LECIF learns a score with strong correlation with the original LECIF score learned from all the features (Pearson correlation of 0.88, Spearman correlation of 0.80) and only a modest reduction in predictive power for aligning regions (AUROC of 0.87 vs. 0.83; AUPRC of 0.21 vs. 0.16). With only 31 mouse features, LECIF learns a score with a weaker correlation with the original score (Pearson correlation of 0.66, Spearman correlation of 0.18) and worse predictive power for aligning regions (AUROC of 0.66; AUPRC of 0.07).

Supplementary Figure 5. Effect of using fewer mouse functional genomic features. To examine the contribution of mouse data to LECIF, we learned two alternative scores using LECIF, one with 10% of the original mouse features and the other with 1% (**Methods**). Specifically, to sample 10% of the mouse features, we randomly selected 6 out of 66 epigenomes in the 15-state ChromHMM chromatin state annotations, selecting 90 chromatin state features. We then additionally sampled 221 features from those corresponding to mouse DNase-seq, ChIP-seq, RNA-seq, and CAGE experiments. To sample 1% of the mouse features, we randomly selected 31 features from those corresponding to mouse DNase-seq, ChIP-seq, RNA-seq, and CAGE experiments. Both scores were learned with all human features originally used in LECIF.

a. Scatter plot showing with a gray dot for each aligning pair of human and mouse regions the LECIF score learned with all features (x-axis) and the alternative score learned with 10% of mouse features. Pearson correlation coefficient (PCC) and Spearman correlation coefficient (SCC) between the two scores are shown in the top left. One hundred thousand pairs of human and mouse regions were randomly selected to be included in the scatter plot.

b. Similar to **a** except showing the alternative score learned with 1% of mouse features in the y-axis.

c. Bar plot showing mean AUROC of the LECIF score learned with all features and the alternative scores learned with all human features and 10% or 1% of mouse features for differentiating aligning pairs from randomly mismatched pairs. One hundred AUROCs were obtained by classifying 100,000 positive and 100,000 negative examples randomly sampled with replacement from all available test examples 100 times, as done in **Fig. 2c**. Mean AUROC is shown above each bar. Standard deviation of the 100 AUROC values was under 0.001 for all scores.

d. Similar to **c** except showing AUPRC instead of AUROC.

We also added the following sentence to the last paragraph in Discussion:

“Applying LECIF to human and mouse with mouse features down-sampled demonstrated that a few hundred annotations from the non-human species may be sufficient to capture a large portion of conservation at the functional genomics level, although the quality of the score will depend on the coverage of the data available for the non-human species.”

Minor points:

- This sentence is difficult to follow due to pronouns in “other species” and “each other”: “For training data, positive examples are pairs of regions between the two species that align at the sequence level while negative examples are randomly mismatched pairs of human and mouse regions that align somewhere in the other species, but not to each other.” It was initially unclear what was meant due to the excessive use of pronouns.

We thank the reviewer for pointing this out. We have revised the sentence to read as follows:

“For training data, positive examples are pairs of human and mouse regions that align at the sequence level while negative examples are randomly mismatched pairs of human and mouse regions that do not align to each other (**Fig. 1a**). All human and mouse regions included in negative examples align somewhere in the mouse and human genomes, respectively, which allows LECIF to learn pairwise characteristics of aligning human and mouse regions instead of the characteristics of regions that align to the other genome in general.”

Reviewer #2 (Remarks to the Author):

Kwon and Ernst developed an interesting machine-learning model to quantitatively define the functional conservation of sequences, between mouse and human. The approach is very solid as they integrated over 1000 datasets, including ChIP-Seq for histone modifications and various TF binding profile. The machine part is well written and detailed. This group has a well-established reputation of develop and release the software to the public (including one of the best known epigenomic software, ChromHMM). Technically I do not have much to add.

We thank the reviewer for the summary and the positive comments.

There is no doubt that the LECIF is important. But what I hope the authors to clarify is: how are we going to use the LECIF score in interpreting biological discoveries or use it in experiment design?

We thank the reviewer for raising this point and apologize that this was unclear. We have edited two sections in the Results and as well as the Discussion to clarify how LECIF can be used in interpreting biological discoveries or designing experiments.

First, we edited the end of section 'LECIF score highlights regions within mouse quantitative trait loci relevant to human disease' to read:

"These results indicate LECIF's potential value in locating regions within mouse QTL that are more likely relevant to a given trait in human."

Second, we edited the end of section 'LECIF score highlights regions with conserved differential methylation patterns linked to phenotype' as follows:

"This supports the potential value of the LECIF score for prioritizing among all loci with epigenetic associations with phenotype in one species the specific loci whose associations are more likely to be shared in the other species."

Lastly, we added the following sentences to the Discussion:

"These results support the potential value of the LECIF score in various applications in the context of model organism research. For example, given a set of phenotypic-associated loci identified in a mouse model, which are increasingly available through efforts like the Mouse Phenome Database, the highest-scoring loci could be prioritized for experimental validation in human cells if possible. Conversely, given human genomic variants or candidate regulatory elements with known associations with a trait, those with the highest LECIF scores could be prioritized for testing in mouse models. In addition, when loci exhibit signals of interest in both species, those with the highest LECIF scores could be prioritized for follow-up experiments."

Does it help us identify tissue-specific or species-specific cis-regulatory elements?

We thank the reviewer for the question. For identifying tissue-specific cis-regulatory elements, we believe it would be better to look at data from each species separately as

the LECIF score is not tissue specific. We note though that LECIF could be useful in identifying elements whose tissue specificity is conserved as we demonstrated that the LECIF score reflects cross-species agreement in H3K27ac experiments manually matched by tissue type in section 'LECIF score highlights human and mouse regions with shared regulatory or transcriptional activity'. LECIF can identify candidate species-specific cis-regulatory regions since low LECIF score may indicate lack of conservation, but we note the LECIF score alone cannot confidently differentiate lack of conservation from lack of evidence based on the available data. To make these points clearer, the paragraph in the Discussion regarding limitations of LECIF now reads (new sentences underlined):

“While we expect LECIF to be useful, we do note a few limitations. LECIF only scores evidence of conservation at the functional genomics level. There thus could be regions that are conserved at the functional genomics level, but have a low LECIF score, since the evidence was not present in the data currently available to LECIF. This makes it difficult to distinguish the case of human-specific regulatory activity from insufficient evidence in the aligning mouse region’s annotations based on a low LECIF score. Fortunately, the interpretation of high LECIF scores is less ambiguous. We also note that the LECIF score’s resolution is limited by the resolution of the input functional genomic annotations and thus does not have the base resolution that sequence-based conservation annotations can have. Additionally, LECIF is designed to aggregate information across multiple tissues and cell types and thus does not provide direct information about a particular tissue.”

Below are some minor technical questions:

1. Can authors split the training site into non-coding and coding? I want to understand where the lower LECIF score genes come from.

We thank the reviewer for the suggestion. Based on the suggestion, we generated two alternative LECIF scores, one learned from a model trained on coding regions and the other on non-coding regions. We compared these scores to the original LECIF score learned from all regions in **Supplementary Fig. 4** also shown below. Within coding regions, the alternative score learned from coding regions was reasonably well-correlated with the original LECIF score with a Pearson and Spearman correlations of 0.71. Within non-coding regions, the score learned from non-coding regions was strongly correlated with the original LECIF score with a Pearson correlation of 0.95 and Spearman correlation of 0.96, which is expected since most training data for the original LECIF score consisted of non-coding regions.

Supplementary Figure 4. Comparison of the LECIF score to scores learned with training data from either non-coding or coding regions.

To evaluate the effect of splitting training examples into coding and non-coding, we learned two separate scores, one from coding examples and the other from non-coding examples (**Methods**). A pair of human and mouse regions was considered coding if the human region overlapped any coding sequence. The same procedure for learning the LECIF score was applied to learn a score from non-coding examples. The same was done for coding examples, except all available training and tuning examples were used for hyperparameter search and then each classifier with optimized parameters was trained on 10,000 positive and 10,000 negative training examples. The scores learned separately on coding and non-coding regions are largely similar to the original LECIF score.

a. Scatter plot showing with a grey dot for a coding region its LECIF score learned from all regions (x-axis) and its score learned from coding regions (y-axis). Pearson correlation coefficient (PCC) and Spearman correlation coefficient (SCC) between the scores are shown in the top left. One hundred thousand pairs of human coding regions were randomly selected to be included in the scatter plot.

b. Similar to **a** except showing with a grey dot for a non-coding region its LECIF score learned from all regions (x-axis) and score learned from non-coding regions (y-axis).

Moreover, based on the reviewer's interest in low-scoring genes, we used chromatin state annotations to characterize low-scoring coding regions based on the LECIF score learned from all regions and the alternative score learned by LECIF from coding regions in **Supplementary Fig. 14** in the revised manuscript. Coding regions that scored low (below 25th percentile among coding regions) based on either score exhibited relatively weak cross-species similarity in chromatin states as expected from low-scoring regions. Specifically, they had Pearson correlation coefficients (PCC) of their chromatin state frequencies between human and mouse epigenomes ranging from 0.1 to 0.5 across different states. In comparison, coding regions that scored high (above 75th percentile among coding regions) based on either score had PCC up to 0.9.

Supplementary Figure 14. Chromatin state similarity in low-scoring coding regions.

As described in **Supplementary Figure 4**, we learned an alternative score using pairs with coding human regions. Here we examine human and mouse chromatin states in low-scoring coding regions based on either the original LECIF score learned from all regions or the alternative score learned from coding regions. Coding regions that score low according to either scores exhibit weak cross-species similarity in their chromatin states as expected.

a. ChromHMM chromatin state annotations in randomly selected pairs that include a human coding region with low LECIF score. The pairs were selected based on whether their human regions overlapped GENCODE annotation of coding sequence (CDS). Each row in the top sub-panel corresponds to a human cell or tissue type. Each row in the bottom sub-panel corresponds to a mouse cell or tissue type. Each column is a randomly selected pair with a human coding region with low LECIF score among all pairs with a human coding region (<25th percentile among coding regions). Each cell shows the color of the chromatin state with which the human or mouse region (column) is annotated in a specific cell or tissue type (row). The chromatin state model and state coloring are the same as in **Fig. 3b** and **Supplementary Fig. 12**. Pairs (columns) were ordered based on hierarchical clustering applied to their chromatin state annotations using Ward’s linkage with optimal leaf ordering⁵.

b. Same as **a**, but with pairs selected based on the alternative score learned from coding regions instead of the LECIF score.

c. Shown for each chromatin state (x-axis) is the state’s PCC in low-scoring pairs with a human coding region based on the LECIF score (<25th percentile among coding regions; bold-colored bars) or the alternative score learned from coding training data (light-colored bars). Each state’s PCC was computed as explained in **Fig. 3b** and **Supplementary Fig. 12** where the correlation is computed between the state’s frequencies in human cell or tissue types and its frequencies in mouse cell or tissue types across all low-scoring pairs restricted to human coding regions.

Based on our analysis, there was no notable difference in spitting the training data into coding and non-coding. We now note the agreement of the scores in the main text as follows in section ‘Comparative evaluation of LECIF’s predictive performance’:

“We also compared the LECIF score to scores learned separately from coding and non-coding genomes and observed that the scores were relatively well-correlated with the original LECIF score in coding (PCC: 0.71) and non-coding (PCC: 0.95) genomes (**Supplementary Fig. 4; Methods**).”

Since our analysis with chromatin states further supports the notion that low-scoring regions show less cross-species agreement in chromatin states than high-scoring regions, we now reference **Supplementary Fig. 14** in section ‘LECIF score highlights human and mouse regions with shared regulatory or transcriptional activity’ as follows:

“Low-scoring pairs of regions were annotated with dissimilar sets of states in human and mouse and the quiescent state more frequently than high-scoring pairs (**Fig. 3b,d, Supplementary Figs. 12-14**).”

In this work, we aimed to learn a genome-wide score that is not specific to any class of DNA elements by leveraging a diverse collection of functional genomic annotations. In general, we do not recommend LECIF for specifically studying gene expression alone as a small fraction of the input functional genomic data to LECIF is directly relevant to gene expression. Methods that specifically model cross-species gene expression such as TROM¹ or Seurat² might be appropriate for comparing gene expression across species. While in principle one could aggregate LECIF scores across gene bodies, different lengths of genes can be a confounder, and thus we feel such analyses would be best left for future work.

2. Page 10 "... In contrast, low-scoring pairs of regions were annotated with dissimilar sets of states in human and mouse and more frequently annotated with the quiescent state than high-scoring pairs ". On the other hand, can authors plot the distribution of score for the unmarked region and all the chromatin states? This can strengthen their claim.

We thank the reviewer for the suggestion. The score distributions of different chromatin states are shown in **Figure 2e** for human and **Supplementary Figure 10** for mouse. We apologize that this was not clear. In the revised version, we moved the score distributions to an earlier figure (**Figure 2** instead of **3**) and also made a stand-alone section titled 'Distribution of LECIF score in chromatin states', which makes this analysis more prominent.

3. It would be interesting to investigate the LECIF scores across TADs boundaries.

We thank the review for the suggestion. We agree this would be interesting to investigate and have now done so using TAD boundaries identified in Dixon et al.³ The analysis is now shown in **Supplementary Fig. 15** in the revised manuscript and also below. Genomic regions overlapping TAD boundaries score higher than the genome average. Moreover, aligning human and mouse regions overlapping TAD boundaries in matched cell types score higher than pairs with either human or mouse region overlapping TAD boundaries in matched cell types. We did not observe these patterns with the human-only baseline score, indicating that LECIF is likely learning the cross-species agreement in TAD boundaries from both human and mouse data.

Supplementary Figure 15. LECIF score and human-only baseline score in topologically associated domain (TAD) boundaries.

a. Box plot showing the distribution of LECIF score of pairs with a human or mouse genomic region overlapping TAD boundaries in human or mouse cell type. Top two cell types listed along the y-axis are human cell types, and the other two are mouse cell types. Each distribution is represented by a boxplot with median (orange solid line), mean (green 'x'), 25th and 75th percentiles (box), and 5th and 95th percentiles (whisker). Orange and green dashed lines vertical lines across the entire panel denote the genome-wide median and mean LECIF scores, respectively.

b. Similar to **a** but showing the distribution of LECIF score of pairs with human and mouse regions with respect to their overlap with TAD boundaries in embryonic stem cells (ESC). Top distribution corresponds to aligning human and mouse regions overlapping TAD boundaries in both hESC and mESC ('Both hESC & mESC'). Second and third distributions correspond to aligning pairs with either human or mouse region overlapping TAD boundaries in ESC ('hESC only' and 'mESC only'). Bottom distribution corresponds to the remaining pairs which are those with neither region overlapping TAD boundaries in ESC.

c-d. Similar to **a-b**, respectively, except for human-only baseline score instead of LECIF score.

These results show that LECIF score is higher at TAD boundaries than average, which are known to be highly conserved between human and mouse, and also higher in conserved TAD boundary regions than in species-specific TAD boundary regions. These patterns are not consistently observed with human-only baseline score.

We have added the following paragraph to section 'LECIF score highlights human and mouse regions with shared regulatory or transcriptional activity' in the results:

"We also investigated the LECIF score at topologically associated domain (TAD) boundaries that were previously identified in human and mouse cell types as they represent an important regulatory genomic feature not provided to LECIF. Human regions overlapping a TAD boundary in any human cell type had a mean LECIF score of 0.17 compared to the genome-wide mean of 0.14 (Mann-Whitney U test $P < 0.0001$). Pairs with human and mouse regions both overlapping a TAD boundary in a matched cell type had an even higher mean of 0.20, scoring significantly higher than pairs with either human or mouse region or neither regions overlapping a TAD boundary in the cell type (**Supplementary Fig. 15**; Mann-Whitney U test $P < 0.0001$)."

Reviewer #3 (Remarks to the Author):

Review of:

Learning a genome-wide score of human-mouse conservation at the functional
genomics level

by Soo Bin Kwon & Jason Ernst

=====
=====

The authors describe a computational approach for assessing the conservation between human and mouse genomic regions, in terms of genome-wide functional genomics datasets. In particular, they present a largely data-driven method that, very importantly, does not rely on an explicit mapping between, or even matching of, underlying datasets between species. This is a conceptual advance over existing approaches, and particularly relevant for current large-scale data generation efforts (e.g. ENCODE), which tend to be sub-optimally coordinated in terms of matching assays between species.

We thank the reviewer for the summary and the positive comments.

Overall, the manuscript is clearly written, albeit a bit dry. I feel that the writing and messaging of the paper can be improved by making the goals of each described analysis, and the problems they address, much more explicit. As it stands, the manuscript reads like most of the "meat" is front-loaded, with more technical (control) analyses in the second half of the paper. Perhaps the manuscript can be shortened in a way that puts full focus on the main message, namely the fully data-driven approach for dealing with the important question of (functional) conservation between species.

We thank the reviewer for this feedback. We have revised the writing of the manuscript based on this advice. In particular, we shortened the technical details of how each downstream analysis was conducted. We also shortened the part on the human-only baseline score since the baseline score is not the focus of this manuscript, summarizing the results instead of reporting every evaluation. Furthermore, we more explicitly specified the purpose of each analysis and possible applications of LECIF throughout the manuscript.

The core statistical and machine learning approaches used are acceptable, and performance evaluations are generally executed according to the latest best-practices.

We thank the reviewer for the positive comment.

That said, there are a number of concerns I have with some of the analyses and the conclusions the authors reach based on these, highlighted in the following points below.

We thank the reviewer for these comments. We have addressed them as described below.

Points:

1.

Page 2, start of first paragraph. It was not clear to me what the exact distinction is here between "pairwise sequence alignment" (regional conservation, allowing for gaps?) and "constrained at the sequence level" (per-basepair conservation?) My understanding is that the current work builds upon the former, but since the method is very much anchored on the presence of some degree of sequence conservation, it is important to make this very clear at this point in the manuscript.

We thank the reviewer for pointing out that this was unclear and apologize for that. The reviewer is correct in drawing the distinction between regional vs. per-basepair conservation. Pairwise sequence alignment considers sequence similarity of large DNA blocks whereas constraint considers sequence similarity at a basepair level in an existing multi-species alignment. Additionally, alignment is defined by being able to confidently determine two loci are homologous, while constraint is called based on observing fewer mutations than expected under a neutral model of evolution. While human and mouse sequences may be similar enough at a region level to be aligned to each other, they may not necessarily have sufficient sequence similarity at the base-resolution to be considered constrained at the sequence level. Both pairwise and multi-species alignments allow for gaps.

We have added the following sentence to the second paragraph of the Introduction to clarify the distinction and the reason why only a small fraction of aligning human and mouse regions are under constraint at the sequence level:

"This is because many bases are within regions whose sequences are similar enough to be aligned between species but not necessarily constrained, which are defined at a higher resolution and generally have even greater sequence similarity."

We also note that there is some increase in sequence constraint in human regions that align to mouse compared to the entire human genome. We now explicitly demonstrate this in **Supplementary Figure 18**, which we reference in the section 'LECIF score's relationship to annotations based on sequence' to explain why we think it is meaningful to compare the LECIF score to sequence constraint annotations. The figure shows that human bases that align to mouse tend to score higher in PhyloP score when compared to the entire genome, but there are still many aligning bases that do not have a high PhyloP score.

Supplementary Figure 18. Distribution of PhyloP score in aligning bases.

Comparison of the distribution of PhyloP score (100 vertebrate) in human genomic bases in general (grey) and bases that align to mouse (red). 1 million bases annotated by PhyloP score were randomly sampled from the genome. Shown in grey is the distribution of PhyloP score of all 1 million bases. Shown in red is the distribution of PhyloP score of bases that align to mouse among the 1 million bases. Twenty equal-width bins ranging from -5 to 8 were used to plot the histogram, covering more than 99% of the score distribution. Bins outside the range are not shown. This comparison demonstrates that although aligning bases have a slightly higher range of sequence constraint than all bases they still have a wide range of constraint.

Also, perhaps I missed this, but what is the starting basis for the fraction of the human genome that can be used with LECIF in this way? 40%?

Human regions that align to mouse cover about 40% of the human genome, to which LECIF is applied. This was previously mentioned in the Introduction, and we now mention this again when we discuss how we applied LECIF to generate genome-wide predictions in section 'Learning evidence of conservation from integrated functional genomic annotations' as follows:

“After training, we used the classifier to make genome-wide predictions of at most 50 bp resolution, annotating the 40% of the human genome that aligns to mouse and those aligning regions in the mouse genome with the LECIF score (Fig. 1b).”

2.

It should be made more clear at an early point in the manuscript what exactly the working resolution of LECIF in this study is (50bp?). Although in the manuscript (page 4), it is mentioned that the /predictions/ are done at every 50bp, it is not made clear (enough) whether this is also the case for the training phase.

We thank the reviewer for pointing out that this was unclear and we apologize for that. The resolution of LECIF, for both training and predictions, is at most 50 bp since we define genomic regions at every 50 bp within each continuous block of aligning bases between human and mouse. These blocks end and begin at every alignment gap in the pairwise alignment. Because the blocks are not necessarily divisible by 50 and

sometimes include frequent alignment gaps, there are many genomic regions shorter than 50 bp. We now mention this earlier in the first paragraph of Results:

“Since neighboring bases are likely annotated by the same annotations and for computational considerations, training examples and predictions were generated at every 50 bp within each pairwise alignment block (**Methods**).”

3.

Page 4, Two million positive and negative examples. Does this mean that the same (two million) genomic regions in each species were re-used for positive (aligning) and negative (non-aligning) sets? And if so, assuming a resolution of 50bp, this training set encompasses 100Mb of the human and mouse genomes?

We thank the reviewer for the questions. Genomic regions used to define positive and negative examples are randomly sampled from the same pool of human and mouse regions. Because of subsampling, a classifier may not necessarily see the exact same set of genomic regions in positive and negative examples. To clarify this, we added the following sentence to section ‘LECIF classifier’ in Methods:

“While the same genomic regions in each species appear in both positive and negative examples given all available training examples, a single neural network may not necessarily encounter the same set of regions in its positive and negative examples due to random sampling.”

As explained in point #2, the coarsest resolution of LECIF is 50 bp, but can be less than 50 bp depending on the size of alignment blocks. To be exact, the training examples cover up to 90 Mb of the human and mouse genomes. We now mention this in the first paragraph of Results:

“As a result, we provided the classifier with more than two million positive and two million negative training examples, which covered up to 90 Mb of the human and mouse genomes.”

4.

Page 7, first paragraph. The authors mention that although LECIF can be used on genomic regions with no sequence conservation between species, doing so would result in a higher false positive rate. I would like the authors to elaborate on this in more detail. In particular, is this because non-conserved regions are less likely to be functional to begin with, or is there a more structural (e.g. sequence composition?) difference between (sequence-based) conserved and non-conserved regions? After all, later the authors mention that LECIF scores are often high when sequence conservation is low ("lacking", see point below).

We thank the reviewer for the questions. First, while LECIF can be used to score any given pair of human and mouse regions in theory, it is difficult to interpret their score if the regions do not align to each other. This is because sequence alignment provides strong prior information when locating candidate pairs of human and mouse regions that may be conserved at the functional genomics level. For example, 8% of aligning pairs

score above 0.5 whereas only 0.1% of randomly mismatched non-aligning pairs score above 0.5. For every aligning pair of human and mouse regions, there are more than 32 million pairs consisting of the same human region and a mouse region that does not align to the human region but somewhere else in the human genome. Based on this and the LECIF score distribution for positive and negative test examples, for every aligning pair scoring above 0.5, there are on average more than 400,000 pairs of non-aligning human and mouse regions that score above 0.5. It is difficult to expect most of these 400,000 pairs to represent true pairs with strong evidence of conservation at the functional genomics level without additional prior evidence from sequence alignment.

Second, LECIF was trained only on human and mouse regions that align to the other species. LECIF is therefore expected to learn patterns in such regions and these patterns may not be frequently found in regions that do not align to the other species. For example, regions that align to the other species tend to overlap genes and regulatory elements whereas other regions “are less likely to be functional to begin with” as the reviewer pointed out.

For better clarity and logical flow, we moved the discussion of false positives to the second paragraph along with the results of this section’s analysis. To elaborate on why we expect more false positives when applying LECIF to non-aligning regions, we revised it to read:

“These results suggested that applying LECIF to non-aligning regions would result in a substantial increase in false positive predictions, which indicates that sequence alignment provides strong prior information in detecting evidence for conservation at the functional genomics level. Moreover, non-aligning regions in general tend to be less conserved and exhibit different properties at the functional genomics level than aligning regions on which LECIF was trained, making LECIF relatively less applicable to such regions.”

5.

Perhaps I missed this in the Methods, but how did the authors deal with unambiguous conservation mappings between human and mouse, e.g. multi-mapping regions?

We thank the reviewer for the question. Given multiple mouse regions that map to a single human region, we chose the mouse region with the highest alignment score. We have added the following sentence to the ‘Pairwise sequence alignment’ sub-section of the Methods for clarification:

“Given multiple mouse genome segments that map to a single human genome segment, we chose the mouse segment with the highest alignment score.”

6.

Figure 1b suggests that predictions are only done on aligning pairs. During model building and evaluation of course also non-aligning pairs need to be considered. Can the authors explain their reasoning for ultimately applying the model only to aligning pairs more clearly?

We thank the reviewer for the question. We are uncertain whether the reviewer is asking about applying the model to (i) pairs of human and mouse regions that do not align to each other but somewhere else in the other species or (ii) pairs of human and mouse regions that do not align to the other species in general.

If the reviewer is asking about the former, we want to clarify that pairs of human and mouse regions that do not align to each other but somewhere else in the other species are provided as negative examples in model building and evaluation. We do make predictions for these pairs to evaluate performance (**Fig. 2c,d**) but we do not explicitly show this step in **Figure 1b**. We apologize that this might not have been clear and now clarify this in the legend of **Figure 1b** as follows:

“Although not shown here, for model evaluation we also generate predictions for randomly mismatched negative pairs held out from training.”

If the reviewer is asking about why LECIF was not applied to human regions that do not align to mouse and mouse regions that do not align to human, please refer to our response to point #8 where we discuss our focus on aligning regions in detail.

Perhaps one thought related to this is what would happen if for instance the human-only baseline model (although clearly not performing as well as the full model) would be run genome-wide.

In modeling, the human-only baseline model does consider regions from the entire human genome regardless of whether they align to mouse because it uses human regions that align to mouse as positive examples and the remaining human regions as negative examples. This information was previously mentioned only in our Methods. We now mention this in the main text as well when we first mention this model as follows:

“We also verified the advantage of integrating human and mouse data by generating a human-only baseline score. The score was learned using human functional genomics data with human regions that align to mouse as positive examples and the rest as negative examples (**Methods**).”

Although the entire human genome is considered in modeling, we generate predictions only in human regions that align to mouse because our purpose with this score is to compare to the LECIF score, which is available for those regions only. If we generated predictions for the entire human genome using this model, it will likely highlight biochemically active regions in a way consistent with how human regions that align differ from the rest of the genome as discussed in our response to point #4.

7.

Weighting of positive and negative examples during classification: why 50x more weight on negative examples? I appreciate the effort the authors have made to compare this scenario to one where positive and negative examples are weighted equally (Supplementary Figure 1), but it remains unclear to me exactly why the 50x weighting is preferable over the alternative. Additionally, this weighting has a profound influence on the LECIF score distribution (see same figure), which begs the question: how was the 50x determined? Why not 10x, 100x?

We thank the reviewer for the question. We decided to weight negative examples more than positive examples because we wanted the LECIF score to highlight regions with strong evidence of conservation. Since the LECIF score reflects evidence of conservation, we are interested in high-scoring regions that show strong evidence and less interested in low-scoring regions since it is not meaningful to focus on lack of evidence. We thus reasoned that it would be helpful to shift the score distribution to be lower such that only a small fraction of the aligning regions score high and attract our attention in the genome browser and downstream analyses. We edited the main text to make this point clearer as follows:

“We weighted negative examples 50 times more than positive examples during training because we wanted the LECIF score to highlight regions with strong evidence of conservation at the functional genomics level. As a result, a small fraction of the aligning regions were highlighted with high LECIF score whereas most aligning regions would have scored high if the score was learned with positive and negative examples weighted equally (**Fig. 2b**).”

We chose 50x as the weight because it shifted the score distribution lower as we wanted. We also manually examined the score in the genome browser to ensure that researchers viewing a DNA segment of interest with the LECIF score as a browser track could easily discern which parts of the segment are highlighted by the score. For example, 80% of the human regions aligning to mouse would have scored higher than 0.5 with a score learned with equal weighting. In contrast, only 8% of the genome scores higher than 0.5 with the LECIF score based on a 50x weighting.

We do recognize that other weights could have been used to achieve a similar effect. Based on the reviewer’s question, we modified **Supplementary Fig. 1** to now include results from additional weighting schemes (10x and 100x), as shown below. We observe that varying weights does not have a substantial effect on the predictive performance of LECIF (AUROC of 0.87 for all weighting schemes) or relative rankings of predictions (Spearman correlation of 0.94 or above). Using different weights is thus not expected to substantially alter our downstream analyses.

Supplementary Figure 1. Effect of different weight ratios between positive and negative examples.

Comparisons of the LECIF score, which was learned with negative examples weighted 50 times more than positive examples, to alternative versions of the score learned with different weighting schemes. To generate each alternative version, we repeated the hyper-parameter search and prediction procedures with the same dataset, but with different weighting scheme.

a,d,g,j. Distribution of a score learned with positive-negative example weight ratio of 1:1, 1:10, 1:50, and 1:100, respectively. Fifty equal-width bins were used to plot this histogram.

b,e,h,k. Scatter plot showing with a gray dot for each aligning pair of human and mouse regions ascore learned with positive and negative examples weighted equally (x-axis) and a score learned with positive-negative example weight ratio of 1:1, 1:10, 1:50, and 1:100, respectively (y-axis). Pearson correlation coefficient (PCC) and Spearman correlation coefficient (SCC) between the two scores are shown in the top left. One hundred thousand pairs of human and mouse regions were randomly selected to be included in the scatter plot.

c,f,i,l. ROC curve of a score learned with positive-negative example weight ratio of 1:1, 1:10, 1:50, and 1:100, respectively, for differentiating positive and negative pairs. Mean ROC curve was obtained by classifying 100,000 positive and 100,000 negative examples randomly sampled with replacement from all

available test examples 100 times Mean area under the ROC curve (AUROC) is shown in the bottom right corner. Standard deviation of the 100 AUROC values was under 0.001 for any weight ratio.

8.

Page 7, first paragraph. The case of regulatory element turnover, i.e. lacking strict (positional) sequence conservation, is an important one. However, I am not convinced that the local translation approach proposed sufficiently takes care of that concern, due to two reasons. One, the 50bp size and non-overlapping nature of the considered neighboring windows are arbitrary with respect to possible turnover events, and are therefore expected to "capture" any potential turnover events with only a small probability. Two, the 50bp resolution allows for plenty of more complex turnover events, not restricted to what is currently considered essentially a linear full-window translation event. I can't help but wonder whether there would be a better way to do this, e.g. by comparing the region-region score to a region-altregion score where altregion is a region (or regions) that contains a proportional increase of sequence conservation at a single-basepair level.

We thank the reviewer for the comment. In response to the reviewer's first point, we note that we cannot detect movements of regulatory elements with a distance smaller than 50 bp because most of the functional genomic annotations provided to LECIF as input features are at a lower resolution than 50 bp. Using smaller and overlapping windows would not resolve this issue. We now mention this in this section as follows:

"We note that because of the resolution at which the LECIF score is defined, even without explicitly expanding the window the score may still be capturing small movements of regulatory sites, which cannot be explicitly detected in the coarse-resolution functional genomics data currently available to LECIF."

In response to the reviewer's second point, we now show additional analysis on how LECIF may detect movements of regulatory elements in certain cases in **Supplementary Fig. 8** as shown below. Specifically, for aligning human and mouse regions with low LECIF score, we observe an improvement in predictive power when we extend the window around the mouse region. If the aligning mouse region exhibited little evidence for conservation at the functional genomics level with the human region, we would be more likely to detect evidence of conservation at an alternative mouse region.

Supplementary Figure 8. Predictive power of region-neighborhood LECIF score for aligning pairs binned by score percentile as a function of neighborhood size around each pair's mouse region.

Similar to **Supplementary Fig. 7a**, except we first bin aligning pairs into five bins based on their region-region LECIF score percentile rank at the aligning regions. For each bin, we evaluate the predictive power

of the region-neighborhood LECIF score of aligning human and mouse regions as a function of neighborhood size. Each line corresponds to a percentile rank bin and is colored based on the color bar on the right. When measuring AUROC, for every positive example falling into a percentile rank bin, we provide a negative example that consists of the same human region of the positive example and a randomly chosen mouse region that aligns somewhere else in the human genome. While extending the neighborhood around each pair's mouse region does not improve predictive power in general, it does help when the aligning regions are scoring low and hard to distinguish from randomly mismatched pairs.

We now reference this analysis in this section as follows:

“We found that as we expanded the window size the predictive power decreased overall. We saw similar results when we repeated the evaluation with pairs stratified by the LECIF score at the aligning regions except for pairs with the lowest LECIF score (**Supplementary Fig. 8**).”

While we briefly presented results for detecting movement of regulatory elements in this section, we decided to focus on aligning pairs of regions in this paper. This is because the section's results showed little to no improvement in predictive power for aligning regions when we included information on proximal non-aligning positions, at least in the way we applied LECIF.

As we mention in our Discussion, we do agree with the reviewer that there could be other strategies in interrogating conservation at the functional genomics level in regions that do not align to each other and modeling movements of regulatory elements. In this paper, given the little prior work on quantifying conservation at the functional genomics level overall, we wanted to keep the method optimized and analyses focused on the more tractable 40% of the human genome that aligns to mouse. We feel the reviewer's suggestion on considering alternative regions with a proportional increase of sequence conservation at a basepair level as well as other possible strategies would be best left to future work.

9.

Page 8, distribution of LECIF scores over input features As it is, this section does not provide much new insight in how the LECIF score behaves. It seems an obvious miss to not mention the clear dichotomy between features associated with (well-conserved) elements like promoters (CAGE, H3K4me3, TSS-states) and (less conserved) elements like enhancers (DNase-seq, ChIP, H3K27ac, enhancer-states). This would drive the point home that LECIF is (on average) pointing towards functionally well conserved elements. All-in-all, I would strongly consider including this as a single sentence in the manuscript, instead of two full paragraphs and a figure.

We thank the reviewer for the feedback. We have significantly shortened this section and highlighted the contrast between well-conserved and less-conserved elements captured by LECIF. However, we prefer to keep the content on chromatin states in the main text because we feel that it provides an effective overview of which DNA elements are highlighted by LECIF and also serves as a validation that LECIF is producing reasonable score distribution for a range of different DNA elements.

10.

The match-analyses described on Page 9 & 10 are more informative, as they are core to the message of the paper that explicit matching of datasets between species may not be necessary. Figure 4 nicely conveys this point, however for panels c & d I would be curious to see what the >95% and <5% LECIF score scenarios would look like for non-aligning pairs.

We thank the reviewer for the positive comment and suggestion. We now show chromatin state annotations for randomly mismatched non-aligning pairs with >95% or <5% LECIF score in **Supplementary Fig. 17**. Even though the human and mouse regions in each pair do not align to each other, high-scoring pairs still show cross-species similarity in chromatin state annotations, more so than low-scoring pairs.

Supplementary Figure 17. Chromatin states in non-aligning pairs with high or low LECIF scores

a. ChromHMM chromatin state annotations in randomly selected pairs of non-aligning human and mouse regions with high LECIF score. The pairs were selected from negative test examples which consist of randomly mismatched pairs of human and mouse regions that do not align to each other (**Methods**). All human and mouse regions included in these pairs do align somewhere in the other species. Each row in the top sub-panel corresponds to a human cell or tissue type. Each row in the bottom sub-panel corresponds to a mouse cell or tissue type. Each column is a randomly selected non-aligning pair with high LECIF score among all non-aligning pairs (>95th percentile). Each cell shows the color of the chromatin state with which the human or mouse region (column) is annotated in a specific cell or tissue type (row). The chromatin state model and state coloring are the same as in **Fig 3b** and **Supplementary Fig. 12**. Pairs (columns) were ordered based on hierarchical clustering applied to their chromatin state annotations using Ward's linkage with optimal leaf ordering.

b. Same as **a**, but with randomly selected non-aligning pairs with low LECIF score (<5th percentile).

c. Shown for each chromatin state (x-axis) is the state's PCC in non-aligning pairs with high (>95th percentile; bold-colored bars) or low (<5th percentile; light-colored bars) LECIF score. Each state's PCC was computed as explained in **Fig. 3b** and **Supplementary Fig. 12** where the correlation is computed between the state's frequencies in human cell or tissue types and its frequencies in mouse cell or tissue types across 100,000 pairs with either high or low LECIF scores. Pairs were randomly sampled from negative test examples as done in **a** and **b**.

An important point in this section, that is in my opinion almost glanced over, is the suggestion that having only human functional genomics data is not sufficient. Perhaps the evidence for this (mostly Supplementary Figure 10) is not strong enough to warrant

a stronger statement on this, but perhaps this analysis can be tightened a bit and this conclusion can be drawn more explicitly.

We thank the reviewer for the comments. In LECIF, using no mouse features would make its positive and negative examples appear identical, providing no predictive information to the classifier. To address this, we had learned the human-only baseline score from human functional genomic data with different definitions of positive and negative examples than LECIF. To more explicitly state that our analysis of this baseline score illustrates the importance of incorporating mouse data, we now edited the paragraph on the human-only baseline to read:

“We also verified the advantage of integrating human and mouse data by generating a human-only baseline score. The score was learned using human functional genomics data with human regions that align to mouse as positive examples and the rest as negative examples (**Methods**). The human-only baseline score was weakly correlated with the human-mouse LECIF score with a PCC of 0.13 and did not reflect cross-species similarity in functional genomic features as strongly as the LECIF score (**Supplementary Figs. 12, 15, 16**). These results support the contribution of mouse data to identifying conserved functional genomic properties.”

We further note that we did include the human-only baseline score in the analysis on partitioning heritability of complex traits. As shown in **Fig. 5b** and **Supplementary Fig. 23**, in general regions that scored high in the LECIF score showed stronger enrichment for heritability than regions that scored high in the human-only baseline score. This further supports the notion that mouse data combined with human data provides more information than human data alone when interpreting phenotype-associated loci.

11.

The question of whether sequence conservation correlates with "functional genomics" conservation is an interesting and important one. The authors address this, in part, by studying the LECIF score distribution relative to overlap with conserved elements and bases called by various computational methods (Figure 5). However, I think the leftover ambiguity mentioned in point 1 above is not helping the authors to make their point. The (combinations of) metrics used (i.e., overlapping elements, conservation methods score strength, number of conserved bases, etcetera) coupled with the weak observed effects (i.e. PCC/SCC, mean/median differences), leaves the reader with a lot of doubt and questions. Can the authors come to a more solid statement based on these results?

We thank the reviewer for the feedback. As mentioned in response to comment #1, we now clarify the distinction between alignment and sequence constraint in the Introduction. We also added the following sentence to this section for further clarification:

“We note that while human regions that align to mouse at the sequence level do show some increase in sequence constraint relative to the entire genome, the majority of aligning regions do not show high levels of sequence constraint (**Supplementary Fig. 18**).”

To more clearly state our interpretation of the moderate correlation between the LECIF score and sequence constraint scores, we edited the end of the first paragraph of this section as follows:

“This moderate correlation may reflect biological difference between sequence conservation and functional genomics conservation, although potentially also the coarse resolution and incompleteness of functional genomics data.”

We note that because of the potential confounders mentioned above, it is difficult to conclusively characterize the relationship between the LECIF score and sequence constraint annotations.

In our opinion, the more informative and meaningful message in this section is that the LECIF score can provide complementary information to sequence constraint annotations about conservation. To strengthen this message, we edited the end of the paragraph that compares the LECIF score to a sequence constraint score using ConsHMM conservation states to read:

“This suggests that the disagreement between the LECIF score and constraint scores could be partly explained by constraint scores not capturing signatures of conservation that are actually present in the multi-species sequence alignment, and further supports that the LECIF score can provide complementary information to sequence constraint scores about conservation.”

We also added the following sentence to the Introduction to make this point early on:

“Overall, we observe that the score can complement sequence conservation annotations in capturing human-mouse conservation and contribute to locating pairs of sequence-aligning regions whose functional genomic properties are likely conserved.”

12.

The second part of this section provides a partial solution by explicitly considering regions that are discordant between sequence conservation and LECIF scores. It is encouraging to see that regions with low sequence conservation (according to phyloP) but high LECIF scores tend to show good inter-species chromatin state agreement. Yet, the authors then mention: "This suggests that the LECIF score can capture conservation at the functional genomics level even in regions that lack sequence constraint, potentially detecting signatures of conservation not captured by sequence constraint annotations." This provides a bit of a conundrum given how LECIF scores are only calculated for regions that already align (i.e. have some level of sequence conservation) between human and mouse. Perhaps the word "lack" is too strong here? Again, this comes back to point 1 above, and clarity of definitions used.

We thank the reviewer for the comments. We hope the clarification on alignment and constraint we now added to the Introduction and this section helps.

We agree that the word “lack” may have been too strong since regions that align between human and mouse may already have some degree of sequence constraint

even if not detected by sequence constraint scores. We therefore changed the above sentence to read:

“This suggests that the LECIF score can capture conservation at the functional genomics level even in regions that align but have limited sequence constraint among aligning regions, potentially detecting signatures of conservation not captured by sequence constraint scores defined from multi-species sequence alignments.”

13.

The authors go on to show association of LECIF score with disease-relevant genetic variation. This was done by performing LD score regression with partitioned heritability (S-LDSC), where one of the included annotations consisted of high-scoring LECIF regions. While the presented enrichment levels are convincing, this does not show how much high-scoring LECIF regions contribute to heritability, controlling for all other annotations. The discussion mentions conditioning on (e.g.) sequence constraint annotation, but the heritability enrichment levels of S-LDSC do not necessarily reflect this. Instead, looking at the coefficients (tau values) of the model to get a clearer view of the contribution specific to LECIF would have been worthwhile.

We thank the reviewer for raising this point. We now report the regression coefficients in **Supplementary Fig. 23** also shown below. For ten out of twelve traits, regions with high LECIF score have positive coefficients, ranging from $2e-8$ to $7e-7$, indicating that they contribute to heritability even when conditioned on existing annotations. We note that both fold enrichment and regression coefficients were computed in the context of existing baseline annotations and thus reflect the contribution of high-scoring region to heritability conditioned on existing annotations. Traits with high fold enrichment level tended to have high regression coefficient as shown in **Supplementary Fig. 23b**.

Supplementary Figure 23. Regression coefficients of regions with high LECIF score in partitioning heritability of complex traits.

As described in the legend of **Fig. 5b**, heritability partitioning was applied to twelve complex traits using a set of annotations which included (i) human regions with high human-mouse LECIF score ($>95^{\text{th}}$

percentile), (ii) regions with high human-only baseline score (>95th percentile), (iii) human regions that align to mouse, and (iv) various baseline annotations used in previous study¹⁴ that includes sequence constraint annotations among others. For 10 out of 12 complex traits, regions with high LECIF score were estimated to have a positive coefficient, suggesting that high-scoring regions contribute to heritability even when conditioned on a large set of existing annotations including sequence constraint annotations.

a. Regression coefficient (tau) values for partitioned heritability of 12 phenotypes in human regions with high LECIF score. Coefficients are shown for human regions with high human-mouse LECIF score (>95th percentile) (blue) and additionally for comparison regions with high human-only baseline score (>95th percentile) (orange) and human regions that align to mouse (gray). Error bars denote jackknife standard error.

b. Scatter plot showing with a grey dot for each complex trait fold enrichment (x-axis) and regression coefficient (y-axis). The regression coefficients and fold enrichment values are the same as those shown in **a** and **Fig. 5b**, respectively. Horizontal and vertical dotted lines correspond to coefficient of 0 and fold enrichment of 1, respectively.

We added the following sentence to this section as well:

“The regression coefficients were largely consistent with the enrichments, with high-scoring regions showing contribution to heritability for several traits even when controlling for the baseline annotation set (**Supplementary Fig. 23**).”

References

1. Yang, Y., Yuan, J., Yang, Y.-C. T., Lu, Z. J. & Li, J. J. Large-scale mapping of mammalian transcriptomes identifies conserved genes associated with different cell states. *Nucleic Acids Res.* **45**, 1657–1672 (2016).
2. Butler, A., Hoffman, P., Smibert, P., Papalexi, E. & Satija, R. Integrating single-cell transcriptomic data across different conditions, technologies, and species. *Nat. Biotechnol.* **36**, 411 (2018).
3. Dixon, J. R. *et al.* Topological Domains in Mammalian Genomes Identified by Analysis of Chromatin Interactions. *Nature* **485**, 376–380 (2012).

Reviewers' Comments:

Reviewer #1:

Remarks to the Author:

The authors have fully addressed our comments. The paper is well-written, timely and impactful.

--

Maxwell Libbrecht, School of Computing Science, Simon Fraser University

Subreviewer: Mariam Arab, School of Computing Science, Simon Fraser University

Reviewer #2:

Remarks to the Author:

The authors have done an excellent job addressing my previous questions. I would recommend the timely publication of this work.

Reviewer #3:

Remarks to the Author:

The writing has much improved, and I appreciate the additional analysis results provided by the authors.

Most of my comments have been addressed satisfactorily, with one item remaining:

In response to point 13, the authors mention the addition of Supplementary Figure 23 with S-LDSC coefficients,

and report that 10 out of 12 coefficients are positive and they thus contribute to trait heritability.

Given the relatively modest coefficient values, the authors should quantify this further by reporting statistical significance of the S-LDSC coefficients, in particular since the fold enrichments they report may not control for the contributions of other annotations.

REVIEWERS' COMMENTS

Reviewer #1 (Remarks to the Author):

The authors have fully addressed our comments. The paper is well-written, timely and impactful.

We thank the reviewer for the positive comments and constructive review.

Reviewer #2 (Remarks to the Author):

The authors have done an excellent job addressing my previous questions. I would recommend the timely publication of this work.

We thank the reviewer for the positive comments and constructive review.

Reviewer #3 (Remarks to the Author):

The writing has much improved, and I appreciate the additional analysis results provided by the authors.

We thank the reviewer for the positive comments and constructive review.

Most of my comments have been addressed satisfactorily, with one item remaining:

In response to point 13, the authors mention the addition of Supplementary Figure 23 with S-LDSC coefficients, and report that 10 out of 12 coefficients are positive and they thus contribute to trait heritability. Given the relatively modest coefficient values, the authors should quantify this further by reporting statistical significance of the S-LDSC coefficients, in particular since the fold enrichments they report may not control for the contributions of other annotations.

We thank the reviewer for raising this point. While the enrichments reported for the LECIF annotation are significant for several traits when including the baseline set of annotations in the model and the coefficients are positive and show a consistent trend with the enrichments, the coefficients are not statistically significant. In general, few coefficients are significant likely due to the various correlations among the input annotations provided to ldsc. Among 60 input annotations, including the LECIF annotation and baseline annotations, about two annotations on average across the 12 traits have statistically significant positive coefficient estimates after multiple testing correction.

As the positive coefficients are not statistically significant, we removed statements from the manuscript that may have implied that to be the case. We have edited the results to focus specifically on the result that regions with high LECIF score enrich for heritability. Based on those revisions, Supplementary Fig. 23 is no longer relevant to the claims made and we have removed it.